# Efficient Multi-agent Offline Coordination via Diffusion-based Trajectory Stitching

**Lei Yuan**[1,2,3]    **Yuqi Bian**[1,2]    **Lihe Li**[1,2]    **Ziqian Zhang**[1,2]    **Cong Guan**[1,2]    **Yang Yu**[1,2,3*]

[1] National Key Laboratory for Novel Software Technology, Nanjing University
[2] School of Artificial Intelligence, Nanjing University    [3] Polixir Technologies
{yuanl,bianyq,lilh,zhangzq,guanc}@lamda.nju.edu.cn, yuy@nju.edu.cn

## Abstract

Learning from offline data without interacting with the environment is a promising way to fully leverage the intelligent decision-making capabilities of multi-agent reinforcement learning (MARL). Previous approaches have primarily focused on developing learning techniques, such as conservative methods tailored to MARL using limited offline data. However, these methods often overlook the temporal relationships across different timesteps and spatial relationships between teammates, resulting in low learning efficiency in imbalanced data scenarios. To comprehensively explore the data structure of MARL and enhance learning efficiency, we propose **M**ulti-**A**gent offline coordination via **Di**ffusion-based **T**rajectory **S**titching (**MADiTS**), a novel diffusion-based data augmentation pipeline that systematically generates trajectories by stitching high-quality coordination segments together. MADiTS first generates trajectory segments using a trained diffusion model, followed by applying a bidirectional dynamics constraint to ensure that the trajectories align with environmental dynamics. Additionally, we develop an offline credit assignment technique to identify and optimize the behavior of underperforming agents in the generated segments. This iterative procedure continues until a satisfactory augmented episode trajectory is generated within the predefined limit or is discarded otherwise. Empirical results on imbalanced datasets of multiple benchmarks demonstrate that MADiTS significantly improves MARL performance.

## 1 Introduction

Multi-agent reinforcement learning (MARL) is a key technology for addressing complex decision-making problems that involve multiple interacting agents (Gronauer & Diepold, 2022). It has demonstrated significant potential in areas such as active voltage control (Wang et al., 2021), large language model (LLM) applications (Sun et al., 2024), and embodied agents (Liu et al., 2024b). Despite the remarkable progress made in MARL, most of its successes are confined to simulated environments, where agents can interact with the environment unlimitedly to gather vast amounts of data for policy improvement. However, this is generally impractical in real-world applications such as autonomous driving and financial transactions, where trial and error can be both costly and risky. This challenge has accelerated research in offline reinforcement learning (RL) (Levine et al., 2020), which focuses on learning from a fixed dataset without interacting with the environment. Numerous efficient methods have been successfully developed (Prudencio et al., 2023), showing promising potential in real-world applications such as industrial process control (Deng et al., 2023), recommender systems (Chen et al., 2024), and legged robot navigation (Weerakoon et al., 2024).

One of the key challenges in offline RL is the issue of distribution shift (Lambert et al., 2022), a phenomenon where unseen state-action pairs are incorrectly estimated. Early works have introduced techniques such as policy constraints (Ran et al., 2023), value function regularization (Mao et al., 2024), uncertainty estimation (Beeson & Montana, 2024), and world model learning (Luo et al., 2024) for data augmentation or direct planning. Recent studies have also leveraged Transformers (Chen et al., 2021) and diffusion models (Janner et al., 2022) to enhance offline RL ef-

---

*Corresponding author.

ficiency from various perspectives (Yang et al., 2023). In addition, while single-agent offline RL has seen rapid growth, most works in offline MARL have primarily focused on adapting successful methods from the single-agent to the multi-agent setting. These approaches have aimed to alleviate extrapolation errors in agent interactions (Yang et al., 2021c), design knowledge distillation mechanisms to bridge the CTDE (Centralized training decentralized execution) gap during policy deployment (Tseng et al., 2022), discover coordination skills from multi-task data (Zhang et al., 2023), and implement efficient policy adaptation strategies (Wu et al., 2024), showing significant progress in multiple scenarios (Formanek et al., 2023; Guan et al., 2024).

Despite recent advances, current offline RL methods are highly correlated with the quality of datasets (Schweighofer et al., 2021), which may suffer from severe performance degradation (Hong et al., 2023) under imbalanced data. Various methods have been developed to address these issues, such as policy regularization (Liu et al., 2024a), data sharing (Yu et al., 2021), efficient data sampling (Hong et al., 2023), and trajectory stitching (Hepburn & Montana, 2022). However, they largely focus on utilizing the provided data, while others aim to learn models from offline data and select actions through planning (Rosete-Beas et al., 2023), or perform data augmentation (He, 2023) to enhance policy learning, showing widespread improvement in sample efficiency. Nevertheless, these methods primarily address single-agent tasks. Multi-agent tasks are significantly more complex due to intricate interactions among agents (Albrecht et al., 2024), which give rise to both temporal relationships across different timesteps and spatial relationships between teammates. Furthermore, in open-environment settings (Yuan et al., 2023), the behavior policy used to collect data might be nonidealized, resulting in severely imbalanced offline data and low learning efficiency. This raises an important question: *Can we augment suboptimal datasets by stitching together high-quality trajectory segments to overcome the temporal and spatial imbalances inherent in multi-agent datasets?*

To comprehensively explore the data structure of multi-agent reinforcement learning (MARL) and improve learning efficiency, we propose **M**ulti-**A**gent offline coordination via **Di**ffusion-based **T**rajectory **S**titching (**MADiTS**), a novel diffusion-based data augmentation pipeline designed to explicitly address imbalances in both temporal and spatial dimensions. Initially, for temporal imbalance, a diffusion model is trained to capture the distribution of trajectory segments and is conditioned on high returns to generate trajectory segments for stitching by head-to-tail concatenation. A bidirectional environmental dynamics constraint is then applied to ensure that only trajectories consistent with the environment's dynamics are selected during the stitching process. Next, for spatial imbalance, an integrated gradient-based method is developed to identify the agents responsible for the suboptimal performance. A partial noising approach is subsequently used to optimize the behaviors of these underperforming agents, leveraging diffusion models to stitch together trajectory segments from various agents. This iterative process continues until a satisfactory augmented trajectory is achieved within the predefined limit or is discarded otherwise, enabling policy optimization with both the generated and original data. Experiments on the imbalanced datasets of on multiple benchmarks including MPE (Lowe et al., 2017), SMAC (Samvelyan et al., 2019), SMACv2 (Ellis et al., 2023), and MAMuJoCo (Peng et al., 2021) demonstrate that MADiTS significantly enhances the performance of Behavior Cloning (BC) and other offline MARL algorithms relative to the original datasets, underscoring MADiTS' effectiveness in tackling the challenges posed by imbalanced data.

## 2 RELATED WORK

Multi-agent reinforcement learning (MARL) (Albrecht et al., 2024) has garnered significant attention recently(Du & Ding, 2021), achieving remarkable success across various complex domains such as active voltage control (Wang et al., 2021) and dynamic algorithm configuration (Xue et al., 2022). A wide range of MARL solutions have been proposed, including value-based approaches like VDN (Sunehag et al., 2018) and QMIX (Rashid et al., 2018), as well as policy gradient methods such as MADDPG (Lowe et al., 2017) and MAPPO (Yu et al., 2022), alongside newer variants like Transformer-based approaches (Wen et al., 2022). However, mainstream MARL methods rely on continuous interaction with the environment to collect data for policy optimization, this challenge has accelerated interest in offline MARL, which focuses on learning policies from pre-collected data. For example, ICQ (Yang et al., 2021c) tackles the extrapolation error by restricting trust to offline data, while MABCQ (Jiang & Lu, 2021) introduces a fully decentralized offline

MARL framework and uses techniques such as value deviation and transfer normalization for efficient learning. OMIGA (Wang et al., 2023) bridges multi-agent value decomposition with policy learning by transforming global-level value regularization into implicit local value regularization. CFCQL (Shao et al., 2023) applies conservative regularization to each agent in a counterfactual manner, then combines them linearly for overall conservative value estimation.

While the aforementioned methods alleviate some challenges, offline RL still faces significant hurdles due to the limited quality and diversity of pre-collected datasets, leading to low learning efficiency. To tackle this issue, various techniques have been proposed (Yu, 2018; Prudencio et al., 2023), such as data sharing (Yu et al., 2021), data augmentation (Yu et al., 2020), knowledge transfer (Bose et al., 2024), and leveraging external knowledge like large language models (LLMs)(Shi et al., 2023). SIT (Tian et al., 2023) further explores offline MARL by explicitly accounting for the diversity of multi-agent trajectories. Among these methods, trajectory stitching has emerged as a promising method of offline trajectory augmentation, synthesizing optimal or near-optimal trajectories from suboptimal ones. For example, MBTS (Hepburn & Montana, 2022) learns a state transition model and value function to stitch together high-quality segments from different trajectories, generating optimal trajectories. BATS (Char et al., 2022) uses an environment model for planning and adds state transitions to fill in missing trajectory segments within offline datasets. More recently, DiffStitch (Li et al., 2024a) leverages the diffusion model (Ho et al., 2020) for trajectory stitching, showing high efficiency in single-agent RL. However, these trajectory stitching methods struggle to effectively learn complex interactions and cooperative behaviors from offline multi-agent trajectory data, as they fail to account for the temporal and spatial imbalances inherent in such data. More about related work could be found in Appendix A.

## 3 BACKGROUND

This paper considers fully cooperative multi-agent task, which is modeled as a Decentralized Partially-Observable Markov Decision Process (Dec-POMDP) (Oliehoek et al., 2016), denoted by a tuple $\mathcal{M} = \langle \mathcal{N}, \mathcal{S}, \mathcal{A}, P, \Omega, O, r, \rho, \gamma \rangle$. Here, $\mathcal{N} = \{1, \cdots, n\}$ is the set of $n$ agents, $\mathcal{S}$ is the global state space, $\mathcal{A} = \mathcal{A}^1 \times \cdots \times \mathcal{A}^n$ is the joint action space of the agents, where $\mathcal{A}^i$ is the action space of agent $i$. $P : \mathcal{S} \times \mathcal{A} \times \mathcal{S} \rightarrow [0, 1]$ is the state transition function. $\Omega$ is the observation space, $O : \mathcal{S} \times \mathcal{N} \rightarrow \Omega$ is the observation function. All agents share the same global reward function $r : \mathcal{S} \times \mathcal{A} \rightarrow \mathbb{R}$. $\rho : \mathcal{S} \rightarrow [0, 1]$ is the initial state distribution. $\gamma \in [0, 1]$ is the discount factor. In each episode, $s_0$ is initially sampled form $\rho(s)$. At each timestep, agent $i$ receives the observation $o^i = O(s, i)$ and outputs an action $a^i \in \mathcal{A}^i$. The joint action $\boldsymbol{a} = (a^1, ..., a^n)$ leads to the next state $s' \sim P(\cdot|s, \boldsymbol{a})$ and a global reward $r(s, \boldsymbol{a})$. For the optimization, all agents aim to learn the joint policy $\boldsymbol{\pi} = \langle \pi_1, \cdots, \pi_n \rangle$, that maximizes the expected discounted return $J(\boldsymbol{\pi}) = \mathbb{E}_{s_0 \sim \rho, a_t^i \sim \pi_i(\cdot|o_t^i), s_{t+1} \sim P(\cdot|s_t, \boldsymbol{a}_t)} \left[ \sum_{t=0}^{T} \gamma^t R(s_t, \boldsymbol{a}_t) \right]$. In our offline setting, agents cannot explore the environment and are trained with a static dataset $\mathcal{D} = \{\tau_i\}_{i=1}^{M}$, where $\tau$ is the joint trajectory composed of transitions $(s_t, \boldsymbol{o}_t, \boldsymbol{a}_t, r_t)_{t=1}^{T}$.

This paper utilizes Denoising Diffusion Probabilistic Models (DDPMs) (Ho et al., 2020) for trajectory generation, which are a class of generative models that allows for sampling from a distribution via iteratively reversing a forward noising process. DDPMs consist of a forward noising process and a reverse denoising process. Given that $\mathbf{x}_0 \sim q(\mathbf{x}_0)$, a forward process gradually adds noise to the data with a pre-defined variance schedule $\{\beta_k\}_{k=1}^{K}$: $q(\mathbf{x}_k|\mathbf{x}_{k-1}) = \mathcal{N}(\mathbf{x}_k; \sqrt{1 - \beta_k}\mathbf{x}_{k-1}, \beta_k \mathbf{I})$. It can be further derived that $p(\mathbf{x}_k|\mathbf{x}_0) = \mathcal{N}(\mathbf{x}_k; \sqrt{\bar{\alpha}_k}\mathbf{x}_0, (1 - \bar{\alpha}_k)\mathbf{I})$, where $\alpha_k = (1 - \beta_k), \bar{\alpha}_k = \prod_{i=1}^{k} \alpha_i$. A reverse denoising process, constructed as $p_\theta(\mathbf{x}_{k-1}|\mathbf{x}_k) = \mathcal{N}(\mathbf{x}_{k-1}; \mu_\theta(\mathbf{x}_k, k), \Sigma_k)$, is then optimized to maximize the evidence lower bound on negative log likelihood defined as $\mathbb{E}_q[\ln \frac{p_\theta(\mathbf{x}_{0:K})}{q(\mathbf{x}_{1:K}|\mathbf{x}_0)}]$. Here $p_\theta(\mathbf{x}_{0:K}) = \mathcal{N}(\mathbf{x}_K; 0, \mathbf{I}) \prod_{k=1}^{K} p_\theta(\mathbf{x}_{k-1}|\mathbf{x}_k)$. Instead of directly training $\mu_\theta$ by optimizing the evidence lower bound, Ho et al. (2020) proposes a simplified surrogate loss:

$$\mathcal{L}_{\text{denoise}}(\theta) = \mathbb{E}_{k, \mathbf{x}_0, \boldsymbol{\epsilon}}[\|\boldsymbol{\epsilon} - \epsilon_\theta(\sqrt{\bar{\alpha}_k}\mathbf{x}_0 + \sqrt{1 - \bar{\alpha}_k}\boldsymbol{\epsilon}, k)\|^2], \tag{1}$$

where $\epsilon_\theta(\mathbf{x}_k, k)$ directly estimates the noise added to produce $\mathbf{x}_k$. In the reverse diffusion process, the samples can be generated by following the recursion: $\mathbf{x}_{k-1} = \frac{1}{\sqrt{\alpha_k}}(\mathbf{x}_k - \frac{1 - \alpha_k}{\sqrt{1 - \bar{\alpha}_k}}\epsilon_\theta(\mathbf{x}_k, k)) + \sqrt{\beta_k}\mathbf{z}$, with $\mathbf{z}$ sampled from $\mathcal{N}(0, \mathbf{I})$. Furthermore, researchers have proposed two types of methods to guide the conditional generation of samples. Classifier-guided methods (Song et al., 2021; Dhariwal & Nichol, 2021) train an additional classifier and add guidance to predicted noise. Classifier-free

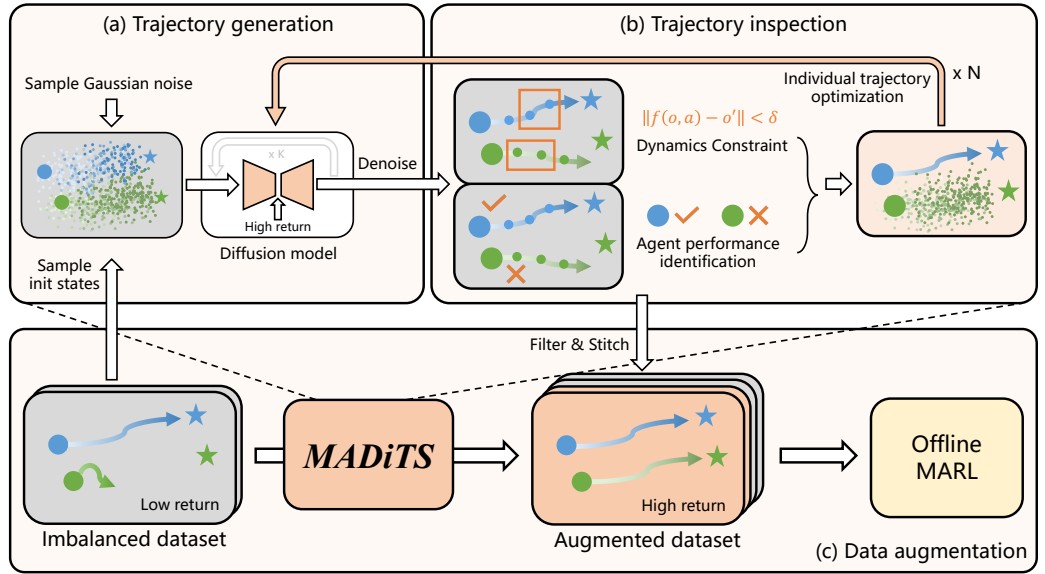

Figure 1: The overall framework of MADiTS.

methods (Ho & Salimans, 2021) modify the original training setup to allow both conditional and unconditional model. The perturbed noise is corrected as $\boldsymbol{\epsilon}_k = \epsilon_\theta(\mathbf{x}, \mathbf{y}, k) + w(\epsilon_\theta(\mathbf{x}, \mathbf{y}, k) - \epsilon_\theta(\mathbf{x}, k))$, where $\mathbf{y}$ is the attribute of label, and $w$ is guidance scale.

## 4 METHOD

In this section, we will introduce the design of Multi-agent offline coordination via Diffusion-based Trajectory Stitching (MADiTS), a novel data augmentation algorithm that systematically generates high-quality coordination from imbalanced dataset (see Figure 1). We first solve the modeling for diffusion in multi-agent scenarios while guaranteeing the bidirectional dynamics consistency in Section 4.1. Next, we propose to overcome the imbalance caused by underperforming agents via integrated gradient in Section 4.2. Finally, an overall pipeline of trajectory stitching for offline MARL within imbalanced dataset is presented in Section 4.3.

### 4.1 TRAJECTORY GENERATION WITH BIDIRECTIONAL DYNAMICS CONSISTENCY

In imbalanced dataset where high-rewarding and low-rewarding states both exist, it is promising to stitch them together by head-to-tail concatenation so that agents can learn to transit to high-rewarding states and achieves the shared goals ultimately. Inspired by the success made by diffusion models in single-agent offline RL (He et al., 2024; Li et al., 2024a), we capture the multimodal distribution of the imbalanced trajectories via a diffusion model $G_\theta$:

$$\max_\theta \mathbb{E}_{(\boldsymbol{o}_t, \boldsymbol{o}_{t+1}, \dots, \boldsymbol{o}_{t+H-1}) \sim \mathcal{D}, t \sim [0, T-H+1]}[\log G_\theta(\boldsymbol{o}_t, \boldsymbol{o}_{t+1}, \dots, \boldsymbol{o}_{t+H-1} | R_t)], \quad (2)$$

where $H$ is the input trajectory length, and $R_t = \sum_{t'=t}^{t+H-1} \gamma^{t'-t} r_{t'}$ is the discounted cumulative return, which can be optimized via the simplified surrogate objective $\mathcal{L}_{\text{denoise}}(\theta)$ defined in Equation 1.

We choose to diffuse over joint observations because actions are more varied, and are less smooth (Ajay et al., 2022). Moreover, it is often multi-discrete in MARL scenarios, making it harder to directly modeling via diffusion model. After training, the trajectory segments are generated following the denoising process:

$$\boldsymbol{o}_t, \hat{\boldsymbol{o}}_{t+1}, \hat{\boldsymbol{o}}_{t+2}, \cdots, \hat{\boldsymbol{o}}_{t+H-1} = G_\theta(\boldsymbol{o}_t, \mathbf{z}_{t+1}, \mathbf{z}_{t+2}, \cdots, \mathbf{z}_{t+H-1} | R_s),$$
$$\text{where } \mathbf{z}_{t'} = (z_{t'}^1, z_{t'}^2, \cdots, z_{t'}^n), \quad \forall t' \in \{t+1, t+2, \cdots, t+H-1\}. \quad (3)$$

Here, $\boldsymbol{o}_k$ denotes the joint observation at time step $t'$ and $\hat{\boldsymbol{o}}_{t'}$ denotes the predicted joint observation at time step $t'$ after denoising. $z_{t'}^i$ refers to the sampled Gaussian noise for all $i \in \{1, 2, \cdots, n\}$,

$R_{\mathrm{s}}$ is a fixed expected return as a conditional input. Furthermore, to infer the action $\boldsymbol{a}_t$ from the generated observation trajectory, we introduce an inverse dynamics model $f_\phi^{\mathrm{inv}}(\boldsymbol{o}_t, \boldsymbol{o}_{t+1}) = \boldsymbol{a}_t$.

Despite the powerful generalization ability of diffusion models, the generated trajectories might violate environment dynamics due to excessively prioritizing high returns. Such inconsistency will accumulate over time, and in turn reduces the reliability of trajectory segments. To mitigate the effects of dynamics inconsistency, we propose a bidirectional dynamics constraint mechanism to identify generated observations that violate dynamics. Specifically, we instantiate another forward dynamics model $f_\psi^{\mathrm{fwd}}(\boldsymbol{o}_t, \boldsymbol{a}_t)$ to predict the next joint observation. Both the inverse and forward dynamics model $f_\phi^{\mathrm{inv}}$ and $f_\psi^{\mathrm{fwd}}$ are implemented with three-layer MLPs (multi-layer perceptrons) , trained using transitions sampled from the offline dataset:

$$\mathcal{L}_{\mathrm{dynamics}}(\phi, \psi) = \mathbb{E}_{(\boldsymbol{o}_t, \boldsymbol{a}_t, \boldsymbol{o}_{t+1}) \sim \mathcal{D}}[\|f_\phi^{\mathrm{inv}}(\boldsymbol{o}_t, \boldsymbol{o}_{t+1}) - \boldsymbol{a}_t\|_2^2 + \|f_\psi^{\mathrm{fwd}}(\boldsymbol{o}_t, \boldsymbol{a}_t) - \boldsymbol{o}_{t+1}\|_2^2]. \quad (4)$$

Once the joint observation trajectory is generated by $G_\theta$ conditioned on the target return, we traverse each observation pair $(\hat{\boldsymbol{o}}_t, \hat{\boldsymbol{o}}_{t+1})$ generated by $G_\theta$, and infer the joint action $\hat{\boldsymbol{a}}_t = f_\phi^{\mathrm{inv}}(\hat{\boldsymbol{o}}_t, \hat{\boldsymbol{o}}_{t+1})$. Then we predict the legal next-step observation $\tilde{\boldsymbol{o}}_{t+1} = f_\psi^{\mathrm{fwd}}(\hat{\boldsymbol{o}}_t, \hat{\boldsymbol{a}}_t)$. We discard the subsequent segment after $\hat{\boldsymbol{o}}_{t+h-1}$ if $\|\hat{\boldsymbol{o}}_{t+h} - \tilde{\boldsymbol{o}}_{t+h}\|$ exceeds a certain threshold $\delta_{\mathrm{recon}}$, which means that the dynamics consistency is severely violated. In practice, we keep the initial joint observation unchanged and restore generated trajectories with bidirectional dynamics consistency $(\boldsymbol{o}_t, \hat{\boldsymbol{a}}_t, \hat{\boldsymbol{o}}_{t+1}, ... \hat{\boldsymbol{o}}_{t+h-1}, \hat{\boldsymbol{a}}_{t+h-1})$ as candidates for data augmentation. Here $2 \le h \le H$ due to the discarded inconsistent segments, the details of trajectory generation will be showed in Section 4.3.

## 4.2 BEHAVIOR CORRECTION OF UNDERPERFORMING INDIVIDUALS

Although the diffusion model with bidirectional dynamics consistency can help generate high-rewarding trajectories that accord with environmental dynamics, the unique spatial imbalance in multi-agent system still makes it difficult to generate trajectory segments with high cooperativeness. Such imbalance is caused due to the underperforming individuals in the system, and it is difficult for the diffusion model to correct their behaviors under such large joint trajectory space.

However, the search space can be largely narrowed down if we have access to the identity of underperforming individuals by fixing the joint trajectory of good ones. This is difficult in a cooperative MARL setting as all the agents share the same reward function, known as credit assignment problem. Inspired by integrated gradient (IG) (Sundararajan et al., 2017), a neural network interpretability method, we can quantify each agent's contribution at each timestep so as to identify underperforming individuals. Given the function $F : \mathbb{R}^d \to \mathbb{R}$, IG computes the attribution of each feature by:

$$\mathrm{IG}_i(x) = (x_i - x_i') \int_{\alpha=0}^{1} \frac{\partial F(x' + \alpha(x - x'))}{\partial x_i} \mathrm{d}\alpha, \quad (5)$$

where $x'$ is the baseline point, and $\frac{\partial F(x)}{\partial x_i}$ is the gradient of $F$ at $x$ with respect to the $i$-th feature. By introducing a path function $\gamma(\alpha) = x' + \alpha(x - x')$ which specifies a path from $x$ to baseline point $x'$, we can define the path integrated gradients (PathIG) as:

$$\mathrm{PathIG}_i(x; \gamma) = \int_{\alpha=0}^{1} \frac{\partial F(\gamma(\alpha))}{\partial \gamma_i(\alpha)} \frac{\partial \gamma_i(\alpha)}{\partial x_i} d\alpha. \quad (6)$$

We first show the team return can be decomposed into the individual contributions of each agents by specifying $F$ of PathIG according to a trainable reward function $g_\omega^{\mathrm{rwd}} : \Omega \times \mathcal{A} \to \mathbb{R}$. Given a joint trajectory segment of length $h$ $\boldsymbol{\tau}_t = (\boldsymbol{o}_t, \boldsymbol{a}_t, \cdots, \boldsymbol{o}_{t+h-1}, \boldsymbol{a}_{t+h-1})$, let $x_t = (\boldsymbol{o}_t, \boldsymbol{a}_t)$, then:

$$\hat{R}(x_t, \boldsymbol{\tau}_t) - r(x_{t+h-1}) = \sum_{i=1}^{n} \sum_{j \in \mathbb{X}_i} \mathrm{PathIG}_j(x_t; \gamma_{\boldsymbol{\tau}_t}), \quad (7)$$

where $\hat{R}(x_t, \boldsymbol{\tau}_t) = \sum_{t'=t}^{t+h-1} r(x_{t'})$ is the return-to-go, $\mathbb{X}_i$ represents the set of observation-action features for agent $i$, $x_{t+h-1}$ is chosen as the baseline point, and $\gamma_{\boldsymbol{\tau}_t}$ is the simple path function starting from $x_t$ to $x_{t+h-1}$. The detailed proof of Equation 7 is provided in Appendix C, and the intuition behind Equation 7 is that the return-to-go can be decomposed into the sum of integrated gradients of each agent's respective features. Thus, we can approximate the contribution of agent

$i$ at timestep $t$ as $\sum_{j \in \mathbb{X}_i} \text{PathIG}_j(x_t; \gamma_{\boldsymbol{\tau}_t})$ provided with the generated joint trajectory $\boldsymbol{\tau}_t$. We first train a team reward prediction model $g_\omega^{\text{rwd}}$ implemented with a three-layer MLP via minimizing:

$$\mathcal{L}_{\text{reward}}(\omega) = \mathbb{E}_{(\boldsymbol{o}_t, \boldsymbol{a}_t, r_t) \sim \mathcal{D}}[\|g_\omega^{\text{rwd}}(\boldsymbol{o}_t, \boldsymbol{a}_t) - r_t\|_2^2]. \tag{8}$$

To comprehensively evaluate each agent's contribution in the generated trajectory segment $(\boldsymbol{o}_t, \hat{\boldsymbol{a}}_t, \cdots, \hat{\boldsymbol{o}}_{t+h-1}, \hat{\boldsymbol{a}}_{t+h-1})$ so that we could find out the underperforming individuals, we first sort the contribution value in an ascent order and derive the ranking $\text{rank}_t^i$. Here, $(\text{rank}_t^1, ..., \text{rank}_t^n)$ is a permutation of $(1, ..., n)$ for any timestep $t$. Thus, the average ranking of each agent's contribution in the joint trajectory segment can be calculated as $\text{rank}_{\text{mean}}^i = \frac{1}{h} \sum_{t'=t}^{t+h-1} \text{rank}_{t'}^i$. For agents whose average ranking value is lower than a certain threshold $\delta_{\text{rank}}$, which means that agent $i$ underperforms in the joint trajectory due to low contribution to the overall team performance, they will be attributed to underperforming individuals. Finally, to correct the behaviors of these underperforming individuals within a smaller trajectory space and improve the cooperativeness, we conduct a resampling process by fixing other agents' generated trajectories:

$$\boldsymbol{o}_t, \hat{\boldsymbol{o}}_{t+1}, \hat{\boldsymbol{o}}_{t+2}, \cdots, \hat{\boldsymbol{o}}_{t+h-1} = G_\theta(\boldsymbol{o}_t, \tilde{\mathbf{z}}_{t+1}, \tilde{\mathbf{z}}_{t+2}, \cdots, \tilde{\mathbf{z}}_{t+h-1} | R_{\text{s}}),$$
$$\text{where } \tilde{\mathbf{z}}_{t'} = (\tilde{z}_{t'}^1, \tilde{z}_{t'}^2, \cdots, \tilde{z}_{t'}^n), \quad \forall t' \in \{t+1, t+2, \cdots, t+h-1\},$$
$$\text{and } \tilde{z}_{t'}^i = \begin{cases} z_{t'}^i, & \text{rank}_{\text{mean}}^i \geq \delta_{\text{rank}} \\ o_{t'}^i, & \text{otherwise.} \end{cases} \tag{9}$$

Here, $\boldsymbol{o}_{t'}$ denotes the joint observation at time step $t'$ and $\hat{\boldsymbol{o}}_{t'}$ denotes the predicted joint observation at time step $t'$ after denoising.

### 4.3 DATA AUGMENTATION PROCESS

We here provide the overall description of the procedure of MADiTS. Given the imbalanced dataset $\mathcal{D}$, we first split the trajectories into segments with length $H$ to formulate $\mathcal{D}_{\text{seg}}$. Additional techniques including return-based filtering and circular shift are applied to facilitate the stitching of high-quality cross-agent individual trajectories (see Appendix D). Afterwards, we train diffusion models $G_\theta$, inverse and forward dynamics model $f_\phi^{\text{inv}}$, $f_\psi^{\text{fwd}}$, and the reward model $g_\omega^{\text{rwd}}$ via objectives defined in Equations 2, 4, and 8, respectively. Details could be found in Appendix E.

To generate a high-rewarding trajectory segment that satisfy both environmental dynamics consistency and cooperativeness, we first sample an joint observation $\boldsymbol{o}_t$ from the dataset $\mathcal{D}$. Then, conditioning the fixed $\boldsymbol{o}_t$ and high return, the diffusion model generates a joint observation segment $(\boldsymbol{o}_t, \hat{\boldsymbol{o}}_{t+1}, \cdots, \hat{\boldsymbol{o}}_{t+H-1})$. We will discard the subsequent part after $\hat{\boldsymbol{o}}_{t+h-1}$ if it violates the environmental dynamics consistency based on the learned inverse and forward dynamics models. To avoid the waste caused by frequent discarding, we will try to re-generate the subsequent segment until the number of failures exceeds a threshold. For each generated trajectory segment that accords with the environmental dynamics, we identify underperforming individuals by calculating the path integrated gradient via the learned reward model, and then correct their behaviors through re-generating the corresponding part in the trajectory segment. It should be noticed that the bidirectional environmental dynamics consistency mechanism will be applied to ensure the legacy of re-generated segment. This iterative procedure of generation and inspection continues until a pre-defined number of trajectories are generated to formulate $\mathcal{D}_{\text{aug}}$. Finally, we augment the original dataset with generated trajectories $\mathcal{D}^* = \mathcal{D} \cup \mathcal{D}_{\text{aug}}$. Since our method MADiTS is algorithm-agnostic, any offline MARL algorithms can be applied and benefit from the augmented dataset. Specifically, we implement our method to the behavior cloning (BC) (Song et al., 2018), OMIGA (Wang et al., 2023) and CFCQL (Shao et al., 2023), and the detailed pseudocode could be found in Appendix F.

## 5 EXPERIMENT

### 5.1 EXPERIMENT SETTINGS

In this section, we introduce (1) the environments to conduct experiments, (2) the mechanism for data collection, (3) the agnostic offline MARL algorithms for evaluation, and the baselines compared with our method, to provide a clear understanding of the experimental setup.

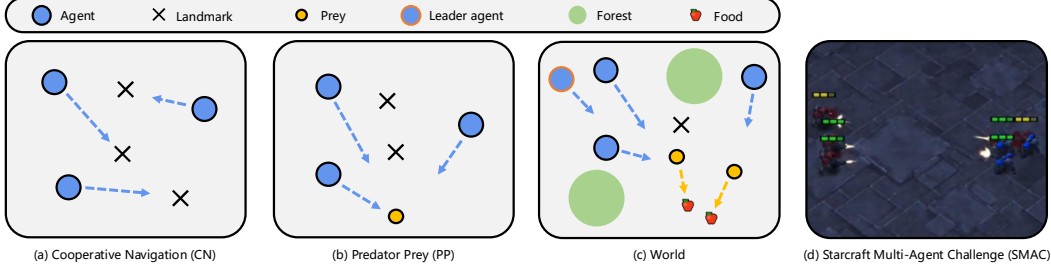

Figure 2: Illustrations of the experimental environments in our work.

**Environments.** We evaluate our method on two widely-used multi-agent benchmarks that require agent cooperation: the Multi-Agent Particle Environment (MPE) (Lowe et al., 2017), the StarCraft Multi-Agent Challenges (SMAC) (Samvelyan et al., 2019), SMACv2 (Ellis et al., 2023), and MA-MuJoCo (Peng et al., 2021). Some of the environments are shown in Figure 2. MPE is a 2D game where particle agents can move, observe each other, and interact with fixed landmarks, including three tasks: Cooperative Navigation (CN), Predator-Prey (PP), and World. In CN, agents must cover different fixed landmarks without colliding with each other. In PP, predator agents collaborate to chase a faster, pre-trained prey agent. The World task introduces more elements including food, forests, and a leader predator based on PP. SMAC consists of a suite of StarCraft II battle scenarios where agents must cooperate to defeat enemy forces controlled by the game's built-in AI. We conduct experiments on maps 3m, 2s3z, 2m_vs_1z, and 12m. SMACv2 is a new version of the benchmark SMAC where scenarios are procedurally generated and add more challenging tasks. We conduct experiments on terran_5_vs_5 and zerg_5_vs_5. MAMuJoCo is a cooperative multi-agent robotic control where multiple agents within a single robot have to solve a task cooperatively. We use 4-agent ant (4ant) configuration in our experiments.

**Datasets.** We construct datasets with varying degrees of imbalance in both temporal and spatial dimensions to evaluate our method. First, behavior policies are trained using MATD3 (Ackermann et al., 2019) on MPE and MAMuJoCo and using QMIX (Rashid et al., 2018) on SMAC and SMACv2, for collecting trajectory data for offline training. Next, during the data collection process, we apply perturbation on the policies to introduce imbalance. For the temporal dimension, at each timestep, there is a small probability $p_{\text{imb}}$ that all agents perform random actions and continue for $t_{\text{imb}}$ steps. For the spatial dimension, perturbations are introduced by randomly selecting certain agents to perform random actions for the entire episode. Two levels of spatial perturbation are designed: moderate (exp-m), where only one agent performs random actions throughout the episode, and severe (exp-s), where the number of random agents is sampled from $\{1, \cdots, n-1\}$. We collect 40k, 20k, 10k, 2k trajectories in MPE, SMAC, SMACv2, MAMuJoCo, respectively.

**Evaluation algorithms and Baselines.** Based on the original dataset, we can apply MADiTS to derive an augmented dataset, and use any agnostic offline MARL algorithms to learn the policies. For a given offline MARL algorithm, the quality of the learned policies can serve as a metric of the augmented dataset's quality. We employ the classic algorithm, behavior cloning (BC), which simply imitates the behavior policy and offers an intuitive reflection of dataset quality, and two state-of-the-art algorithms, OMIGA (Wang et al., 2023) and Counterfactual Conservative Q-Learning (CFCQL) (Shao et al., 2023). Additionally, to evaluate the effectiveness of MADiTS, we first compare it with a simple baseline "Original", which directly utilizes the original perturbed datasets for policy learning. Then, we include MA-MBTS, by extending the well-developed data augmentation method, MBTS (Hepburn & Montana, 2022), into multi-agent settings. MBTS is a model-based trajectory stitching method that generates new actions to connect high-quality segments from different trajectories. In MA-MBTS, individual observations and actions are replaced by joint ones to perform stitching. MADiff(Zhu et al., 2023a) is a diffusion model-based offline MARL algorithm designed to predict future joint observations for decision-making. We directly apply stitching to MADiff for data augmentation. We also include another strong baseline where offline MARL algorithms directly learns from balanced datasets, denoted as "Balanced", which are collected by the same behavior policies without adding any perturbation. Details of these algorithms can be found in Appendix G.

Table 1: Evaluation results on multi-agent imbalanced datasets. The mean and standard error are computed based on the normalized average return or average battle won rate of the evaluation algorithms trained on the datasets, with 5 different random seeds. We **bold** the highest scores on exp-m and exp-s datasets, respectively. The results of exp-m and exp-s on 2m_vs_1z are the same since levels of spatial perturbation for environments of 2 agents are the same. Results of augmentation on balanced datasets can be found in Appendix H.

| Envs | Algs | Balanced | Original | | MA-MBTS | | MADiff | | MADiTS (Ours) | |
|---|---|---|---|---|---|---|---|---|---|---|
| | | exp | exp-m | exp-s | exp-m | exp-s | exp-m | exp-s | exp-m | exp-s |
| CN | BC | 45.22 ± 8.60 | 17.27 ± 3.66 | 9.03 ± 3.40 | 40.34 ± 6.89 | 35.77 ± 6.47 | 40.44 ± 3.75 | 30.24 ± 4.82 | **43.44 ± 6.11** | **37.82 ± 4.45** |
| | OMIGA | 55.53 ± 10.68 | -2.27 ± 57.07 | -11.60 ± 58.90 | 3.39 ± 49.17 | -12.86 ± 78.01 | 19.26 ± 67.68 | 7.27 ± 66.89 | **23.02 ± 69.69** | **22.91 ± 44.94** |
| | CFCQL | 54.07 ± 10.10 | -33.40 ± 44.28 | -56.44 ± 40.38 | 7.62 ± 25.38 | -27.06 ± 23.15 | 30.08 ± 12.97 | 21.88 ± 10.71 | **39.57 ± 16.14** | **28.60 ± 20.94** |
| PP | BC | 52.77 ± 6.15 | 48.49 ± 4.69 | 49.21 ± 3.90 | 46.77 ± 5.89 | 48.06 ± 3.34 | 49.55 ± 6.50 | 49.93 ± 3.74 | **54.85 ± 4.23** | **55.50 ± 4.28** |
| | OMIGA | 58.34 ± 3.57 | 37.90 ± 25.34 | 26.73 ± 40.67 | 39.93 ± 18.02 | 60.88 ± 2.27 | 47.48 ± 20.97 | 58.44 ± 4.76 | **63.02 ± 3.40** | **63.71 ± 5.67** |
| | CFCQL | 51.02 ± 6.76 | 45.03 ± 4.62 | 28.88 ± 6.29 | 44.64 ± 11.74 | 29.17 ± 8.41 | 45.50 ± 6.76 | 30.04 ± 7.02 | **47.38 ± 3.74** | **32.25 ± 10.98** |
| World | BC | 54.00 ± 5.18 | 47.29 ± 3.00 | 48.30 ± 5.96 | 52.02 ± 7.32 | 49.29 ± 5.12 | 50.79 ± 2.62 | 51.55 ± 4.96 | **54.25 ± 4.35** | **52.84 ± 5.29** |
| | OMIGA | 56.90 ± 5.94 | 54.23 ± 6.20 | 52.92 ± 5.64 | 42.99 ± 23.45 | 40.83 ± 17.34 | 49.55 ± 24.99 | 48.26 ± 24.75 | **57.35 ± 5.89** | **58.59 ± 9.32** |
| | CFCQL | 50.70 ± 6.15 | 28.59 ± 1.90 | 18.14 ± 18.16 | 28.48 ± 6.05 | 31.63 ± 10.37 | 28.63 ± 10.14 | 34.35 ± 6.91 | **29.62 ± 10.91** | **39.36 ± 1.91** |
| **Average** | | 53.17 | 27.01 | 18.35 | 34.02 | 28.41 | 40.14 | 36.88 | **45.83** | **43.50** |
| 2m_vs_1z | BC | 0.16 ± 0.31 | 0.03 ± 0.06 | 0.03 ± 0.06 | 0.32 ± 0.23 | 0.32 ± 0.23 | 0.30 ± 0.18 | 0.30 ± 0.18 | **0.35 ± 0.20** | **0.35 ± 0.20** |
| | OMIGA | 0.93 ± 0.15 | 0.59 ± 0.28 | 0.59 ± 0.28 | 0.65 ± 0.20 | 0.65 ± 0.20 | 0.97 ± 0.04 | 0.97 ± 0.04 | **0.98 ± 0.02** | **0.98 ± 0.02** |
| | CFCQL | 0.97 ± 0.03 | 0.72 ± 0.24 | 0.72 ± 0.24 | 0.89 ± 0.08 | 0.89 ± 0.08 | 0.92 ± 0.05 | 0.92 ± 0.05 | **0.94 ± 0.05** | **0.94 ± 0.05** |
| 3m | BC | 1.00 ± 0.00 | 0.42 ± 0.05 | 0.38 ± 0.20 | 0.26 ± 0.33 | 0.28 ± 0.19 | 0.44 ± 0.18 | 0.36 ± 0.15 | **0.51 ± 0.19** | **0.43 ± 0.18** |
| | OMIGA | 0.97 ± 0.02 | 0.93 ± 0.05 | 0.90 ± 0.03 | 0.98 ± 0.00 | 0.94 ± 0.04 | **1.00 ± 0.00** | 0.94 ± 0.03 | **1.00 ± 0.00** | **0.96 ± 0.03** |
| | CFCQL | 0.95 ± 0.03 | 0.90 ± 0.06 | 0.80 ± 0.08 | 0.87 ± 0.05 | 0.77 ± 0.18 | 0.89 ± 0.05 | 0.81 ± 0.09 | **0.94 ± 0.07** | **0.89 ± 0.06** |
| 2s3z | BC | 0.92 ± 0.05 | 0.73 ± 0.11 | 0.68 ± 0.14 | 0.73 ± 0.29 | 0.67 ± 0.09 | 0.75 ± 0.06 | 0.67 ± 0.05 | **0.78 ± 0.06** | **0.69 ± 0.26** |
| | OMIGA | 0.97 ± 0.03 | 0.86 ± 0.16 | 0.67 ± 0.05 | 0.81 ± 0.05 | 0.68 ± 0.13 | **1.00 ± 0.00** | 0.71 ± 0.15 | **1.00 ± 0.00** | **0.76 ± 0.19** |
| | CFCQL | 0.96 ± 0.02 | 0.73 ± 0.15 | 0.66 ± 0.14 | 0.76 ± 0.13 | 0.67 ± 0.13 | 0.80 ± 0.09 | 0.63 ± 0.15 | **0.92 ± 0.02** | **0.68 ± 0.26** |
| **Average** | | 0.87 | 0.65 | 0.60 | 0.69 | 0.65 | 0.78 | 0.70 | **0.82** | **0.74** |

## 5.2 PERFORMANCE COMPARISON

In this section, we analyze MADiTS's effectiveness on augmenting various kinds of offline MARL datasets, and eventually enhancing policy learning.

Specifically, we apply MADiTS and the baseline method to augment the given pre-collected datasets. Offline MARL algorithms are then deployed on the augmented datasets and extra balanced datasets to learn multi-agent policies. The performance of various learned policies are presented in Table 1, where we record the normalized average return and battle won rate for MPE and SMAC tasks, respectively. From the results, we can first observe that Original suffers significant performance drop compared with Balanced across all datasets and offline MARL algorithms, showing the necessity for data augmentation techniques in such imbalanced situations. However, simply utilizing such techniques designed for single-agent settings, MA-MBTS shows only marginal improvement compared with Original. It indicates that multi-agent coordination scenarios require specifically designed techniques to handle both temporal and spatial imbalance. MADiff shows obvious improvement compared to the Original, showcasing the necessity of using diffusion to model data distributions. But it does not account for dynamics consistency or data imbalance, which results in a noticeable gap in performance compared to MADiTS on imbalanced datasets. On the contrary, our method MADiTS achieves the best performance on all imbalanced datasets and offline MARL algorithms, and a comparable average results compared with Balanced. Notably, in some environments such as PP and 2m_vs_1z, The performance of MADiTS even exceeds policies learned from balanced datasets, highlighting that MADiTS successfully leverages the high-quality segments within the collected trajectories to further improve the quality of the datasets.

To thoroughly evaluate the general effectiveness of MADiTS, we utilize the MPE balanced datasets constructed by Pan et al. (2022), which are collected using behavior policies of varying qualities, and evaluate the above data augmentation techniques. Similarly, MA-MBTS demonstrates only slight improvement over Original, while MADiTS consistently outperforms both. These results prove the versatility and generality of MADiTS on various offline settings. Detailed configurations and results can be found in Appendix H.2.

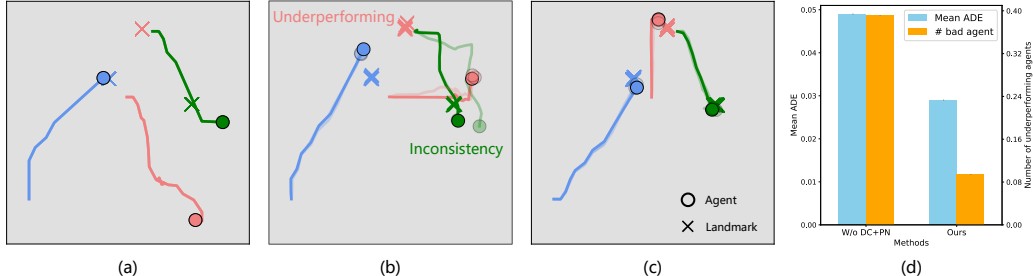

Figure 3: Visualization of different trajectories in CN. Lighter-colored trajectories are the ones extracted from observations of other agents. (a) An original trajectory in imbalanced dataset (b) A trajectory generated by w/o DC+PN. (c) A trajectory generated by MADiTS. (d) Mean ADE of entity trajectories from different agents' observations, and the average number of underperforming agents in datasets augmented by w/o DC+PN and MADiTS.

### 5.3 ILLUSTRATIVE EXAMPLES OF STITCHED TRAJECTORIES

To further investigate how MADiTS addresses temporal and spatial imbalances and improves the quality of imbalanced datasets, we visualize trajectories generated by MADiTS and its variant without dynamics constraint (DC) and partial noising (PN), denoted as **w/o DC+PN**. Specifically, we first render a pre-collected imbalanced trajectory in the original CN exp-m dataset in Figure 3(a), where the agent colored in red fails to cover a landmark. Given the initial observations of this trajectory, w/o DC+PN generates an improved trajectory where the red agent moves closer to the landmarks, as shown in Figure 3(b). Nevertheless, on one hand, the new trajectory suffers from dynamics inconsistency, as the individual trajectory of the agent colored in dark green differs from the one observed by the teammates (light green). On the other hand, agents still have a certain distance from the landmarks, indicating the generated trajectory is suboptimal. After applying both DC and PN, our method successfully generates an optimal trajectory with consistent dynamics, as shown in Figure 3(c), which will be added to the dataset to enhance policy learning.

We also statistically analyze 1000 randomly sampled episodes from the exp-m dataset. Figure 3(d) displays the mean ADE (average displacement error), which measures the average deviation of entities from the observations of all agents, and the average number of underperforming agents per episode. It is evident that MADiTS reduces the mean ADE by almost half compared with w/o DC+PN, indicating that it generates a new trajectory that adheres more to environmental dynamics. Additionally, in the original imbalanced dataset, the number of random agents is set to 1. In the stitching-generated dataset, we consider the number of underperforming agents in a trajectory as the number of uncovered landmarks at the end of the episode. This value drops from 0.391 of w/o DC+PN to 0.094 of our method, demonstrating that PN significantly improves the quality of generated trajectories. How each module of MADiTS influences the coordination ability of the learned policies could be found in the next section.

### 5.4 ABLATION AND SENSITIVITY STUDIES

In this section, we first investigate the effectiveness of each module in MADiTS. We conduct ablation studies by applying BC on the exp-m dataset from the MPE benchmark after data augmentation. Specifically, we test 3 variants: (1) **w/o DC**: removes the bidirectional environmental dynamics constraint mechanism, treating all trajectory segments generated by the diffusion model as compliant with environmental dynamics, (2) **w/o PN**: removes partial noising, not addressing any spatial imbalance between agents, and (3) **w/o DC+PN**: removes both of the aforementioned modules. As shown in Figure 4(a), we can find that the average returns of all the variants significantly exceed that of the original dataset, confirming the necessity of data augmentation for imbalanced datasets. When the dynamics constraint is removed, w/o DC suffers from performance degradation in all MPE tasks, indicating that generated trajectories with dynamics inconsistency can be harmful to policy learning. Furthermore, the variant w/o PN also demonstrates a decrease in performance, showing that iterative individual trajectory optimization effectively improves data quality. Removing both

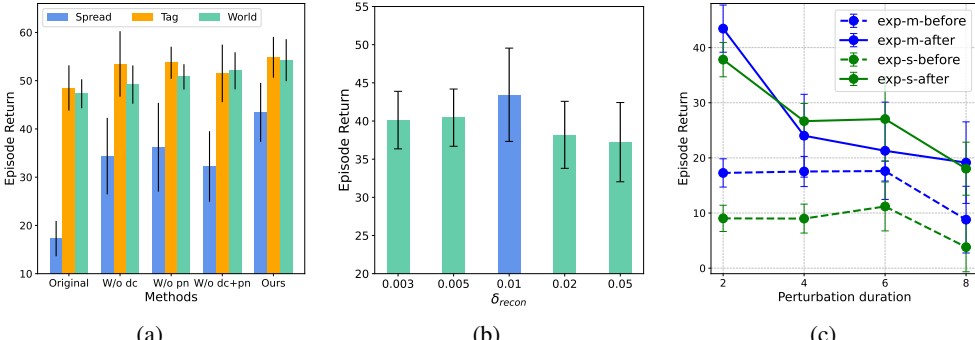

Figure 4: (a) Ablation results on exp-m datasets in MPE. (b) The impact of reconstruction threshold $\delta_{\text{recon}}$. (c) results on CN in various perturbation settings. All the results are computed by the average return of BC policies trained on datasets with 5 different random seeds.

modules, w/o DC+PN achieves worse performance in most tasks, emphasizing their indispensable roles in MADiTS. The full method consistently outperforms other variants in all tasks, illustrating that the designed modules within MADiTS can complement each other and contribute to the effective data augmentation.

Next, as MADiTS includes multiple hyperparameters, we conduct experiments to examine their sensitivity. One of the most important hyperparameters is the reconstruction threshold $\delta_{\text{recon}}$, which controls the strictness of the dynamics constraint. If it's too small, stitching a complete trajectory would take an unacceptable amount of time. On the other hand, if too large, overly lenient constraint will fail to filter out low-quality trajectories. By grid search, we find that $\delta_{\text{recon}} = 0.01$ is the best choice in CN as shown in Figure 4(b). More results on other hyperparameters, like augmentation ratio $r_{\text{aug}}$ and regeneration limit $l_{\text{limit}}$, could be found in Appendix H.3.

## 5.5 EVALUATION IN ADDITIONAL PERTURBATION SETTINGS

Finally, we further explore MADiTS's robustness against various perturbation settings. Specifically, we test the data augmentation effects under varying severity of imbalance in the CN tasks, by setting different duration of temporal imbalance, denoted as $t_{\text{imb}}$, to $\{2, 4, 6, 8\}$. Using the average return of the BC policy in the online environment as the quality metric, we assess the changes in dataset quality before and after stitching, as shown in Figure 4(c). The results demonstrate that MADiTS consistently enhances data quality, even with more severe imbalanced datasets, proving its robustness and generalization. In addition to fully trained expert policies, we also collect datasets by applying perturbations to medium-performing policies, and similar results about quality improvement could be found in Appendix H.4.

## 6 CLOSING REMARKS

In this work, we introduce MADiTS, a novel diffusion-based data augmentation pipeline that significantly enhances the performance of offline MARL algorithms, particularly on datasets with temporal and spatial imbalances. Empirical evaluations on imbalanced datasets across multiple benchmarks highlight the effectiveness of MADiTS in improving dataset quality. The method operates under the assumption that the quality of an agent's behavior over time is reflected in its contribution to team returns, with high returns serving as the criterion for the diffusion model to capture its distribution. However, this assumption may not hold in environments with extremely sparse rewards. Additionally, in extreme scenarios where the diffusion model lacks exposure to high-quality cooperative segments during training, generating Out-Of-Distribution (OOD) segments becomes a challenging problem in data generation. Incorporating external knowledge, such as Large Language Models (LLMs) (Sun et al., 2024), offers a promising avenue for addressing this issue. Future work will aim to overcome these limitations and expand the method's applicability to more complex multi-agent systems, such as those involving multiple embodied agents (Liu et al., 2024b).

## ACKNOWLEDGMENTS

This work is supported by the National Science Foundation of China(62495093, U24A20324), the Natural Science Foundation of Jiangsu (BK2024119, BK20243039). We thank Tencent AI Arena for their support, and the anonymous reviewers for their support and helpful discussions on improving the paper.

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

## A   MORE DETAILS ABOUT RELATED WORK

**Multi-agent Reinforcement Learning (MARL)** has made significant progress, making it well-suited for addressing large-scale, complex, real-time, and uncertain real-world problems. Modeling such problems as single-agent systems is often inefficient and fails to align with real-world conditions. In contrast, formulating them as Multi-Agent Systems (MAS)(Albrecht et al., 2024) is

typically more appropriate. Moreover, when agents share a common objective, the problem becomes a cooperative MARL task(Oroojlooy & Hajinezhad, 2022), which has seen substantial advancements across diverse domains such as pathfinding (Sartoretti et al., 2019), active voltage control (Wang et al., 2021), dynamic algorithm configuration (Xue et al., 2022), and Large Language Model (LLM) applications (Liu et al., 2024b). Numerous approaches have been developed to enhance agent coordination, including policy-based methods like MADDPG (Lowe et al., 2017) and MAPPO (Yu et al., 2022), value-based techniques such as VDN (Sunehag et al., 2018) and QMIX (Rashid et al., 2018), as well as approaches leveraging transformer architectures (Wen et al., 2022). These methods have demonstrated impressive coordination capabilities across various tasks, including SMAC, Hanabi, and GRF (Yu et al., 2022). Beyond these approaches and their variants, other methods have been proposed to explore cooperative MARL, such as efficient communication polices (Zhu et al., 2022), offline policy deployment (Zhang et al., 2023), model learning in MARL (Wang et al., 2022), robustness to perturbations (Guo et al., 2022), sparse reward environments (Li et al., 2024b), ad hoc teamwork (Wang et al., 2024a), and training paradigms like CTDE (Centralized Training with Decentralized Execution) (Lyu et al., 2021). Additionally, several methods have been introduced to design testbeds for MARL algorithms, such as SMAC(Samvelyan et al., 2019), offline data environments (Formanek et al., 2023), and communication testbeds (Guan et al., 2024), etc.

**Offline Multi-agent Reinforcement Learning**  Offline reinforcement learning (RL)(Levine et al., 2020; Prudencio et al., 2023) has garnered significant research interest recently, focusing on a data-driven training paradigm that eliminates the need for direct interaction with the environment(Prudencio et al., 2023). Early works (Fujimoto et al., 2019) primarily addressed the challenges of distributional shift in offline learning, focusing on behavior-constrained policies to reduce extrapolation errors when estimating unseen data (Kumar et al., 2020). Offline Multi-Agent Reinforcement Learning (MARL) is an emerging and promising area of research (Zhang et al., 2021a). One class of offline MARL approaches focuses on learning policies from offline data with policy constraints. For example, ICQ (Yang et al., 2021c) effectively mitigated extrapolation errors by leveraging only offline data in MARL scenarios. MABCQ (Jiang & Lu, 2021) introduced a fully decentralized setting for offline MARL, employing techniques like value bias and transfer normalization for more efficient learning. OMAR (Pan et al., 2022) combined first-order policy gradients with zero-order optimization to circumvent local optima. MADT (Meng et al., 2023b) utilized the sequential modeling power of transformers, applying it to both offline and online MARL tasks. Research (Tian et al., 2023) further explored offline MARL by explicitly accounting for the diversity of agent trajectories and proposed a new framework called Shared Individual Trajectories (SIT). In another work (Tseng et al., 2022), a teacher policy is first trained using full observation, action, and reward data, after which student policies are distilled from the teacher policy, capturing structural relationships between the teacher and agent behaviors. ODIS (Zhang et al., 2023) introduced a novel algorithm for offline MARL, focusing on discovering cooperative skills from multi-task data. Recently, the Off-the-Grid MARL (OG-MARL) framework was released (Formanek et al., 2023), offering a new benchmark for offline MARL dataset generation and algorithm evaluation. M3 (Meng et al., 2023a) innovatively proposed multi-task and multi-agent offline pre-training modules to learn higher-level transferable policy representations. Lastly, OMAC (Wang & Zhan, 2023) introduced an offline MARL algorithm based on coupled value decomposition, decomposing global value functions into local and shared components while ensuring consistent credit assignment between global state values and Q-value functions.

**Sample Efficient Reinforcement Learning**  tackles the challenge of minimizing the number of interactions an agent requires with its environment to learn an optimal policy. In traditional reinforcement learning (RL), agents often need extensive exploration, which can be costly or impractical in real-world applications like robotics, autonomous driving, and healthcare, where data collection is expensive or risky. The goal of sample-efficient RL is to maximize learning performance while minimizing environmental interactions (Yu, 2018). In single-agent RL, common techniques include image-based transformations such as cropping, rotation, and flipping, which are particularly effective in visual tasks, as demonstrated by RAD (Laskin et al., 2020) and DrQ (Yarats et al., 2021). Other approaches, such as perturbing or resampling from the experience replay buffer, encourage exploration. Model-based methods like Dreamer (Hafner et al., 2019) and MOPO (Yu et al., 2020) leverage environmental dynamics models to generate synthetic data. In multi-agent reinforcement learning (MARL), the complexity of agent interactions presents unique challenges for data augmentation, as improper methods can lead to system non-stationarity. Some approaches

address this by exploiting the structure of multi-agent systems, utilizing symmetry (van der Pol et al., 2021), policy similarity (Yang et al., 2021b), parameter sharing (Li et al., 2024c), or learning a meta-policy for multi-task environments (Zhang et al., 2021b). Inspired by the rapid development of foundation models (Yang et al., 2023), recent work has begun to leverage diffusion models (Zhu et al., 2023a; Li et al., 2023; Zhu et al., 2023b), Transformers (Wen et al., 2022), and large language models (LLMs) for MARL (Zhang et al., 2024), showing improvements in some settings. While these approaches contribute to sample efficiency, no current methods focus on generating trajectories and performing trajectory stitching for enhanced policy learning.

## B   MORE COMPARISONS WITH OTHER OFFLINE MARL METHODS

Our method focuses on addressing sample efficiency issues by leveraging diffusion models to perform trajectory stitching for data augmentation in temporally and spatially imbalanced datasets. This approach allows offline MARL algorithms to achieve better performance by learning from the enhanced dataset. In this section, we outline other prominent methods in the field of offline MARL and compare their differences with MADiTS.

SIT (Tian et al., 2023) is an offline MARL method that learns an effective joint policy from agent-wise imbalanced multi-agent datasets, addressing spatial imbalance in particular. It uses an attention-based reward decomposition network to perform offline credit assignment, identifying high-quality individual trajectories for sharing among agents. Additionally, it employs a graph attention network for conservative policy training. Unlike SIT, which directly learns a joint policy from imbalanced datasets, our approach focuses on addressing sample efficiency issues and introduces a data-oriented augmentation pipeline. MADiTS employs a diffusion model-based trajectory stitching mechanism to enhance dataset quality. The augmented datasets can be used by any agnostic offline MARL algorithm, offering flexibility and modularity.

DOM2 (Li et al., 2023) is a diffusion model-based offline MARL algorithm designed to improve policy expressiveness and diversity. It achieves significant gains in performance, generalization, and data efficiency by employing an accelerated solver for diffusion-based policy construction and a policy regularizer, while also scaling up the dataset size to enhance policy learning. DOM2 demonstrates outstanding results on balanced datasets, outperforming traditional conservatism-based offline MARL methods. In contrast, MADiTS focuses on addressing the complex data structures inherent in imbalanced datasets. By tackling temporal and spatial imbalances, MADiTS enhances dataset quality, providing better support for other offline MARL algorithms and improving overall performance in imbalanced scenarios.

MADiff (Zhu et al., 2023a) is another diffusion model-based offline MARL algorithm, designed to predict future joint actions for decision-making by modeling teammates. It employs an attention-based diffusion model to capture joint observation sequences and infers actions for planning. MADiff achieves superior performance on balanced datasets, excelling in standard offline MARL settings. By contrast, MADiTS specifically targets the data imbalance problem in offline MARL, addressing sample efficiency issues. While MADiff improves learning efficiency in balanced settings, MADiTS innovatively focuses on sample efficiency issues for imbalanced datasets. To emphasize the differences, we compare the performance of MADiff extended to data augmentation with MADiTS across various environments. The results demonstrate the superior effectiveness of MADiTS in addressing data imbalance challenges.

## C   THEORETICAL PROOF

Before presenting the proof of Equation 7, we first introduce Integrated Gradient (IG) (Sundararajan et al., 2017), which is an interpretability technique for neural networks, initially proposed in the field of computer vision. IG leverages path integrals of gradients along a path between a baseline input and the actual input to summarize how each feature affects the output of the deep neural network as the model's prediction moves from $F(b)$ to $F(x)$ along the line connecting the $b$ and $x$.

Formally, given an input $x$, IG computes the attribution of each feature $i$ by accumulating the gradients of function $F$ with respect to feature $i$, using a straight line as the integration path, defined by $\gamma(\alpha) = x' + \alpha(x - x')$, where $x'$ is the baseline and $x$ is the input. Then Integrated Gradient for

feature $i$ is then given by:

$$\text{IG}_i(x) = (x_i - x_i') \int_{\alpha=0}^{1} \frac{\partial F(x' + \alpha(x - x'))}{\partial x_i} \, d\alpha, \tag{10}$$

where $\frac{\partial F(x)}{\partial x_i}$ represents the gradient of function $F$ at point $x$ with respect to the $i$-th feature.

**Lemma 1.** *(Sundararajan et al., 2017) If $F : \mathbb{R}^d \to \mathbb{R}$ is differentiable almost everywhere then*

$$\sum_j IG_j(x; \gamma_\tau) = F(x) - F(b), \tag{11}$$

*where $IG_j(x; \gamma_\tau)$ represents the integrated gradient of function $F$ along the path $\tau$ w.r.t the $j$-th dimension of the input $x$ over a straight-line path and $b$ is the baseline for the straight-line path.*

Lemma 1 shows how to compute the straight-line path integral when the function is almost everywhere differentiable. Due to its excellent interpretability, IG is also used in reinforcement learning. For instance, RUDDER (Arjona-Medina et al., 2019) applies integrated gradient to reinforcement learning, addressing the problem of sparse delayed rewards in single-agent reinforcement learning and demonstrating excellent performance. By applying IG, researchers show:

**Lemma 2.** *(Yang et al., 2020) For any joint observation-action trajectory segment $\boldsymbol{\tau}_t = (\boldsymbol{o}_t, \boldsymbol{a}_t, \cdots, \boldsymbol{o}_{t+h-1}, \boldsymbol{a}_{t+h-1})$ from timestep $t$ to timestep $t + h - 1$, we have:*

$$\sum_{j \in \mathbb{X}_i} PathIG_j(x_t; \gamma_{\boldsymbol{\tau}_t}) = \sum_{t'=t}^{T-1} \sum_{j \in \mathbb{X}_i} IG_j(x_{t'}; \gamma_{\boldsymbol{\tau}_{t'}^{t'+1}}). \tag{12}$$

As is shown: the trajectory from timestep $t$ to timestep $t + h - 1$ is split according to the timestep, that is, $\boldsymbol{\tau}_t = (\boldsymbol{\tau}_t^{t+1}, \boldsymbol{\tau}_{t+1}^{t+2}, \cdots, \boldsymbol{\tau}_{t+h-2}^{t+h-1})$, where $\boldsymbol{\tau}_t^{t+1}$ represents the trajectory from $(\boldsymbol{o}_t, \boldsymbol{a}_t)$ to $(\boldsymbol{o}_{t+1}, \boldsymbol{a}_{t+1})$. The integral of the entire path segment can be obtained by summing the straight-line paths of each adjacent timestep. Lemma 2 shows how to calculate the path integral on the trajectory path in the field of reinforcement learning.

Based on Lemma 1 and Lemma 2, the proof of Equation 7 can be given below:

*Proof.* According to the definition of return-to-go $\hat{R}(x_t, \boldsymbol{\tau}_t) = \sum_{t'=t}^{t+h-1} r(x_{t'})$:

$$\begin{aligned}
\hat{R}(x_t, \boldsymbol{\tau}_t) - r(x_{t+h-1}) =& \hat{R}(x_t, \boldsymbol{\tau}_t) - \hat{R}(x_{t+1}, \boldsymbol{\tau}_t) + \hat{R}(x_{t+1}, \boldsymbol{\tau}_t) - \hat{R}(x_{t+2}, \boldsymbol{\tau}_t) \\
&+ \cdots + \hat{R}(x_{t+h-2}, \boldsymbol{\tau}_t) - \hat{R}(x_{t+h-1}, \boldsymbol{\tau}_t).
\end{aligned} \tag{13}$$

For a given $\boldsymbol{o}_t$ and $\boldsymbol{a}_t$, the denoising process generates $\boldsymbol{\tau}_t$ conditioned on a fixed expected return that adheres to a predefined distribution, there exists a randomized function $\tilde{R}$ such that the reward function can be expressed as $r(x_t) = \tilde{R}(x_t) - \tilde{R}(x_{t+1})$. By Lemma 1, we have

$$\begin{aligned}
\hat{R}(x_t, \boldsymbol{\tau}_t) - r(x_{t+h-1}) &= \tilde{R}(x_t) - \tilde{R}(x_{t+1}) + \tilde{R}(x_{t+1}) - \tilde{R}(x_{t+1}) + \cdots + \tilde{R}(x_{t+h-2}) - \tilde{R}(x_{t+h-1}) \\
&= \sum_{j \in x} \text{IG}_j(x; \gamma_{\boldsymbol{\tau}_t^{t+1}}) + \sum_{j \in x} \text{IG}_j(x; \gamma_{\boldsymbol{\tau}_{t+1}^{t+2}}) + \cdots + \sum_{j \in x} \text{IG}_j(x; \gamma_{\boldsymbol{\tau}_{t+h-2}^{t+h-1}}) \\
&= \sum_{j \in x} \left( \text{IG}_j(x; \gamma_{\boldsymbol{\tau}_t^{t+1}}) + \text{IG}_j(x; \gamma_{\boldsymbol{\tau}_{t+1}^{t+2}}) + \cdots + \text{IG}_j(x; \gamma_{\boldsymbol{\tau}_{t+h-2}^{t+h-1}}) \right)
\end{aligned} \tag{14}$$

By Lemma 2, we have

$$\begin{aligned}
\hat{R}(x_t, \boldsymbol{\tau}_t) - r(x_{t+h-1}) &= \sum_{j \in x} \left( \text{IG}_j(x; \gamma_{\boldsymbol{\tau}_t^{t+1}}) + \text{IG}_j(x; \gamma_{\boldsymbol{\tau}_{t+1}^{t+2}}) + \cdots + \text{IG}_j(x; \gamma_{\boldsymbol{\tau}_{t+h-2}^{t+h-1}}) \right) \\
&= \text{PathIG}_1(x; \gamma_{\boldsymbol{\tau}_t^{t+h-1}}) + \text{PathIG}_2(x; \gamma_{\boldsymbol{\tau}_t^{t+h-1}}) + \cdots + \text{PathIG}_d(x; \gamma_{\boldsymbol{\tau}_t^{t+h-1}}).
\end{aligned} \tag{15}$$

Classify $1, 2, \cdots, d$ according to the agents they belong to, we have:

$$
\begin{aligned}
\hat{R}(x_t, \boldsymbol{\tau}_t) - r(x_{t+h-1}) &= \text{PathIG}_1(x; \gamma_{\boldsymbol{\tau}_t^{t+h-1}}) + \text{PathIG}_2(x; \gamma_{\boldsymbol{\tau}_t^{t+h-1}}) + \cdots + \text{PathIG}_d(x; \gamma_{\boldsymbol{\tau}_t^{t+h-1}}) \\
&= \sum_{j \in \mathbb{X}_1} \text{PathIG}_{x_j}(x; \gamma_{\boldsymbol{\tau}_t^{t+h-1}}) + \sum_{j \in \mathbb{X}_2} \text{PathIG}_{x_j}(x; \gamma_{\boldsymbol{\tau}_t^{t+h-1}}) \\
&\quad + \cdots + \sum_{j \in \mathbb{X}_n} \text{PathIG}_{x_j}(x; \gamma_{\boldsymbol{\tau}_t^{t+h-1}}) \\
&= \sum_{i=1}^{n} \sum_{j \in \mathbb{X}_i} \text{PathIG}_j(x; \gamma_{\boldsymbol{\tau}_t^{t+h-1}}).
\end{aligned}
$$

(16)

$\square$

This equation shows that the return-to-go can be decomposed into the sum of integrated gradients of each agent's respective features, providing possibility of offline credit assignment between agents.

## D  CIRCULAR SHIFT

To facilitate cross-agent integration of excellent trajectories by allowing excellent individual behaviors of one agent to be learned by other homogeneous agents, we propose circular shift method to allows the excellent individual behaviors of one agent to be uniformly distributed across all identical agent positions. Therefore, we define the cyclic shift operator $C : \mathcal{T}^H \to \mathcal{T}^H$ such that:

$$
C \left( \begin{bmatrix} o_t^1 & a_t^1 & \cdots & o_{t+H-1}^1 & a_{t+H-1}^1 \\ o_t^2 & a_t^2 & \cdots & o_{t+H-1}^2 & a_{t+H-1}^2 \\ \vdots & \vdots & \ddots & \vdots & \vdots \\ o_t^n & a_t^n & \cdots & o_{t+H-1}^n & a_{t+H-1}^n \end{bmatrix} \right) = \begin{bmatrix} o_t^n & a_t^n & \cdots & o_{t+H-1}^n & a_{t+H-1}^n \\ o_t^1 & a_t^1 & \cdots & o_{t+H-1}^1 & a_{t+H-1}^1 \\ \vdots & \vdots & \ddots & \vdots & \vdots \\ o_t^{n-1} & a_t^{n-1} & \cdots & o_{t+H-1}^{n-1} & a_{t+H-1}^{n-1} \end{bmatrix} \quad (17)
$$

Specifically, we first compare the cumulative discounted rewards of different trajectory segments at the same timestep and select the top $r_{\text{cir}}\%$ of the joint observation-action trajectory segments to form the set $\mathcal{D}_{\text{good\_seg}}$. For all trajectories in this set, assuming all agents are homogeneous, we apply the circular shift operator $n-1$ times to each trajectory, adding each newly generated trajectory segment to the original dataset:

$$
\mathcal{D}_{\text{seg}} \leftarrow \mathcal{D}_{\text{seg}} \cup \{C^k(\boldsymbol{\tau}) \mid \boldsymbol{\tau} \in \mathcal{D}_{\text{good\_seg}}, 1 \le k \le n-1, k \in \mathbb{N}\}. \quad (18)
$$

Here, $\mathcal{D}_{\text{seg}}$ denotes the dataset of trajectory segments obtained by partitioning the original dataset $\mathcal{D}$ into segments of length $H$. For hetergeneous settings, circular shift is conducted within groups of agents of the same state and action space. By applying circular shift, we can increase the proportion of excellent trajectory segments in the original imbalanced dataset, which facilitates the cross-agent stitching of excellent trajectories and thus improves the optimization effect for suboptimal agents.

## E  IMPLEMENTATION DETAILS AND HYPERPARAMETERS

We introduce the model implementation details of generative model $G_\theta$, inverse dynamics model $f_\phi^{\text{inv}}$, the forward dynamics model $f_\psi^{\text{fwd}}$, and the team reward prediction model $g_\omega^{\text{rwd}}$. The implementation of offline MARL algorithms and values of hyperparameters are provided subsequently.

Specifically, we use MADiff (Zhu et al., 2023a) for the implementation of our diffusion model, where a U-Net is used to model the individual trajectories of agents and an attention layer is applied before every decoder block in the U-Net of all agents. Additionally, three-layer multi-layer perceptrons are used to implement the inverse dynamics model, forward dynamics model, and team reward prediction model, respectively, with ReLU used as the activation function between layers.

For the training stages of these models, the loss function is defined as:

$$
\begin{aligned}
\mathcal{L}(\theta, \phi, \psi, \omega) &= \mathbb{E}_{k \sim U(k), \mathbf{x}_0 \sim \mathcal{D}_{\text{seg}}, \beta \sim \text{Bern}(p)} \left[ \| \epsilon - \epsilon_\theta(\mathbf{x}_k, (1-\beta)R(\mathbf{x}_0) + \beta \emptyset, k) \|^2 \right] \\
&\quad + \alpha_{\text{dynamics}} \mathcal{L}_{\text{dynamics}}(\phi, \psi) + \alpha_{\text{reward}} \mathcal{L}_{\text{reward}}(\omega),
\end{aligned}
$$

(19)

where $\mathbf{x}_0 = (\boldsymbol{o}_1, \boldsymbol{o}_2, \cdots, \boldsymbol{o}_H)$ is the observation sequence sampled from trajectory segments dataset $\mathcal{D}_{\text{seg}}$m, $k$ is sampled from a uniform distribution $U(k)$, $p$ is the dropout rate for conditional diffusion model, $\mathcal{L}_{\text{dynamics}}$ and $\mathcal{L}_{\text{reward}}$ are losses in Equation 4 and 8, $\alpha_{\text{dynamics}}$ and $\alpha_{\text{reward}}$ are hyperparameters.

For the implementation of offline MARL algorithms for evaluation, we use behavior cloning, OMIGA (Wang et al., 2023) and CFCQL (Shao et al., 2023) for evaluating the quality of datasets before and after data augmentation. Specifically, we use the 'Offline MARL framework - OffPyMARL' codebase (Zhang, 2023) [1]and default hyperparameters from GitHub to implement these algorithms, with training steps set to 100k.

The experiments were conducted on servers outfitted with GeForce RTX 2080 Ti. We compare the computational costs of MADiTS and other data augmentation method on the Cooperative Navigation task. For MADiTS, the model training phase takes about 36 GPU hours of a single GeForce RTX 2080 Ti and the trajectory stitching phase takes about 4 GPU hours. As comparison, MADiff takes about 36 GPU hours on the same device and MA-MBTS takes about 48 hours for looking for valid transitions for stitching. We can see that MADiTS achieves better performance under comparable computational costs. including MBTS, MADiff, and MADiTS.

The normalized scores in Table 1 and 4 are calculated as $S_{\text{norm}} = 100 \times (S - S_{\text{floor}})/(S_{\text{ceil}} - S_{\text{floor}})$, where $S_{\text{ceil}}$ and $S_{\text{floor}}$ represent the ceiling and floor scores for each task, determined by the deciles of episode returns in the expert datasets. Specifically, for the tasks Spread, Tag, World, and 4ant, the respective ceiling and floor scores are $\{1592, 971\}$, $\{801, 320\}$, $\{400, 182\}$, and $\{5976, 1033\}$.

The hyperparameters used in our work can be categorized into two groups: environment-specific shown in Table 2 and environment-independent shown in Table 3.

Table 2: Environment-independent hyperparameters

| Hyperparameter | Value | Hyperparameter | Value |
|:---:|:---:|:---:|:---:|
| $r_{\text{cir}}$ | 10% | $f_\omega^{\text{rwd}}$ hidden dim | 256 |
| $r_{\text{aug}}$ | 1:1 | lr | 2e-4 |
| $\omega$ | 1.2 | training steps | 1e6 |
| $K$ | 200 | optimizer | Adam |
| condition dropout rate | 0.25 | batch size for stitching | 256 |
| batch size for training | 32 | $l_{\text{limit}}$ | 3 |
| $H$ | 8 | $\alpha_{\text{dynamics}}$ | 10 |
| $f_\phi^{\text{inv}}$ hidden dim | 256 | $\alpha_{\text{reward}}$ | 0.01 |
| $f_\psi^{\text{fwd}}$ hidden dim | 256 | IG step size | 10 |

Table 3: Environment-independent hyperparameters

| Hyperparameters | CN | PP | World | 2m_vs_1z | 3m |
|:---:|:---:|:---:|:---:|:---:|:---:|
| $\delta_{\text{recon}}$ | 0.01 | 0.03 | 0.02 | 0.2 | 0.2 |
| $l_{\text{tot\_limit}}$ | 5 | 5 | 5 | 10 | 10 |
| $\delta_{\text{rank}}$ | 2.33 | 2.33 | 2.33 | 1.5 | 2.33 |
| $R_{\text{s}}$ | 1800 | 1000 | 500 | 20 | 20 |
| **Hyperparameters** | **2s3z** | **12m** | **terran_5_vs_5** | **zerg_5_vs_5** | **4ant** |
| $\delta_{\text{recon}}$ | 5 | 5 | 5 | 5 | 2 |
| $l_{\text{tot\_limit}}$ | 10 | 20 | 20 | 20 | 30 |
| $\delta_{\text{rank}}$ | 2.33 | 8 | 2.33 | 2.333 | 2.33 |
| $R_{\text{s}}$ | 20 | 20 | 20 | 20 | 4000 |

---

[1]https://github.com/zzq-bot/offline-marl-framework-offpymarl

## F Pseudocode

---

**Algorithm 1** Data Augmentation Process

---

**Input**: Original dataset $\mathcal{D}$, generative model $G_\theta$, inverse and forward dynamics models $f_\phi^{\text{inv}}$, $f_\psi^{\text{fwd}}$, team reward prediction model $g_\omega^{\text{rwd}}$.

1: Train generative model $G_\theta$, inverse dynamics model $f_\phi^{\text{inv}}$, forward dynamics model $f_\psi^{\text{fwd}}$, team reward prediction model $g_\omega^{\text{rwd}}$ according to Algorithm 2.
2: $\mathcal{D}_{\text{aug}} = \emptyset$
3: **while** $|\mathcal{D}_{\text{aug}}| \leq r_{\text{aug}} \cdot |\mathcal{D}|$ **do**
4:     Sample an initial joint observation $o_1$ from $\mathcal{D}$ as the current observation.
5:     Initialize $\tau_{\text{gen}} = \{o_1\}$, total regeneration count $\text{regen}_{\text{tot}} = 0$, single regeneration count $\text{regen}_{\text{single}} = 0$.
6:     Using $\{o_1\}$ as the initial joint observation, a joint trajectory $\tau_{\text{gen}}$ is generated according to Algorithm 3.
7:     $\mathcal{D}_{\text{aug}} \leftarrow \mathcal{D}_{\text{aug}} \cup \{\tau_{\text{gen}}\}$
8: **end while**
9: $\mathcal{D}^* \leftarrow \mathcal{D} \cup \mathcal{D}_{\text{aug}}$
10: return $\mathcal{D}^*$

---

**Algorithm 2** Model Training Process

---

**Input**: Original dataset $\mathcal{D}$, generative model $G_\theta$, inverse and forward dynamics models $f_\phi^{\text{inv}}$, $f_\psi^{\text{fwd}}$, team reward prediction model $g_\omega^{\text{rwd}}$.

1: Split $\mathcal{D}$ into trajectory segments $\mathcal{D}_{\text{seg}}$ of length $H$, and find the top $r_{\text{cir}}\%$ joint observation-action trajectory segments $\mathcal{D}_{\text{good\_seg}}$ based on the return ranking.
2: Apply circular shift described in Appendix D: $\mathcal{D}_{\text{seg}} \leftarrow \mathcal{D}_{\text{seg}} \cup \{C^k(\tau) \mid \tau \in \mathcal{D}_{\text{good\_seg}}, 1 \leq k \leq n - 1, k \in \mathbb{N}\}$.
3: Train diffusion model $G_\theta$ based on Equation 2.
4: Train inverse dynamics model $f_\phi^{\text{inv}}$, forward dynamics model $f_\psi^{\text{fwd}}$, and team reward prediction model $g_\omega^{\text{rwd}}$ based on Equation 4 and 8.

---

The data augmentation process (Algorithm 1) contains conducting model training first (Algorithm 2) and generate trajectories by stitching (Algorithm 3). During the model training process, we first identify high-quality trajectory segments from the original dataset and apply circular shift (Lines 1-2). Then we train the diffusion model and other models required for the augmentation process (Lines 3-4). During data augmentation (Algorithm 1), line 1 uses the diffusion model and other models obtained from the training process, and then lines 2 to 9 generate a certain number of joint trajectories to expand the original dataset.

The process of generating a single trajectory is detailed in Algorithm 3. We generate a trajectory segment first (Lines 2-3), and apply dynamics constraint, truncating the trajectory to retain only the parts that pass the checks (Lines 4-13). Then we discard trajectory segments that consistently fail to meet the constraints after multiple generation attempts (Lines 14-16). If we identify any underperforming agents (Lines 17-20), we will optimize their behavior by partial noising (Line 21). Finally, we apply dynamics constraint again to the optimized joint trajectories, truncating them to retain only the valid parts (Lines 22-32).

## G Detailed Description of the Evaluation Algorithms and Baselines

We provide a more detailed introduction to the evaluation algorithms and baselines of the experiment in this section.

**OMIGA** (Wang et al., 2023). OMIGA provides a structured approach to transform global value regularization into local value regularization, which supports efficient in-sample learning. This

---

**Algorithm 3** Single Trajectory Generation Process

---

**Input**: Original dataset $\mathcal{D}$, generative model $G_\theta$, inverse dynamics model $f_\phi^{\text{inv}}$, forward dynamics $f_\psi^{\text{fwd}}$, team reward prediction model $g_\omega^{\text{rwd}}$.

1: **while** A trajectory generation is not finished **do**
2:     Initialize the valid trajectory segment $\tau_{\text{valid}} = \emptyset$.
3:     Condition on the current observation $o_t$ to generate joint observation trajectory segment $\tau_{\text{obs}}$ of length $H$ according to Equation 3.
4:     **for** $k = t, t+1, \cdots, t+H-2$ **do**
5:         Detect whether $(\hat{o}_k, \hat{o}_{k+1})$ in $\tau_{\text{obs}}$ satisfies the dynamics consistency, and obtain the joint action $\hat{a}_k$ and the team reward $r_k$.
6:         **if** Valid **then**
7:             $\tau_{\text{valid}} \leftarrow \tau_{\text{valid}} \cup \{\hat{a}_k, r_k, \hat{o}_{k+1}\}$.
8:             $\text{regen}_{\text{single}} = 0$.
9:         **else**
10:            $o_t = o_k$
11:            $\text{regen}_{\text{tot}} \leftarrow \text{regen}_{\text{tot}} + 1$, $\text{regen}_{\text{single}} \leftarrow \text{regen}_{\text{single}} + 1$.
12:         **end if**
13:     **end for**
14:     **if** $\text{regen}_{\text{tot}} \geq l_{\text{tot\_limit}}$ or $\text{regen}_{\text{single}} \geq l_{\text{limit}}$ **then**
15:         **BREAK**
16:     **end if**
17:     Inspect underperforming agents $\mathcal{B} = \mathscr{B}(\tau_{\text{valid}})$.
18:     **if** $\mathcal{B} = \emptyset$ **then**
19:         $\tau_{\text{gen}} = \tau_{\text{gen}} \cup \tau_{\text{valid}}$
20:     **else**
21:         Use the diffusion model resampling to obtain $\tau_{\text{regen\_obs}}$ described in Equation 9.
22:         **for** $k = t, t+1, \cdots, t+h-2$ **do**
23:             Detect whether $(\hat{o}_k, \hat{o}_{k+1})$ in $\tau_{\text{regen\_obs}}$ satisfies the dynamics consistency, and obtain the joint action $\hat{a}_k$ and the team reward $r_k$.
24:             **if** Valid **then**
25:                 $\tau_{\text{gen}} \leftarrow \tau_{\text{gen}} \cup \{\hat{a}_k, r_k, \hat{o}_{k+1}\}$.
26:             **else**
27:                 $o_t = o_k$
28:             **end if**
29:         **end for**
30:     **end if**
31: **end while**
32: return $\tau_{\text{gen}}$.

---

method bridges the gap between multi-agent value decomposition and policy learning by integrating offline regularization techniques, facilitating principled multi-agent reinforcement learning.

**CFCQL** (Shao et al., 2023). CFCQL computes conservative value estimates for each agent in a counterfactual manner and then aggregates them into an overall conservative value function. This approach avoids treating the multi-agent system as a single high-dimensional entity and improves upon directly applying single-agent methods to multi-agent settings.

**MBTS** (Hepburn & Montana, 2022). MBTS is a model-based trajectory stitching technique that learns a state transition model and value function. By connecting high-quality trajectory segments, MBTS generates new trajectories that replace suboptimal data, effectively enhancing multi-agent performance through state-action exploration and improved dataset composition.

**MADiff** (Zhu et al., 2023a). MADiff is another diffusion-based offline MARL algorithm, designed to predict future joint observations for decision-making by modeling teammates. It employs an attention-based diffusion model to capture joint observation sequences and infers actions for planning. In our experiments, we compare the performance of MADiff extended to data augmentation with MADiTS across various environments.

Table 4: Evaluation results on additional multi-agent imbalanced datasets. The mean and standard error are computed based on the normalized average return or average battle won rate of the evaluation algorithms trained on the datasets, with 5 different random seeds. We **bold** the highest scores on exp-m and exp-s datasets, respectively.

| Envs | Algs | Balanced | Original | | MA-MBTS | | MADiff | | MADiTS (Ours) | |
|------|------|----------|----------|--------|---------|--------|--------|--------|---------------|--------|
| | | exp | exp-m | exp-s | exp-m | exp-s | exp-m | exp-s | exp-m | exp-s |
| 12m | BC | 0.95 ± 0.03 | 0.83 ± 0.11 | 0.63 ± 0.28 | 0.81 ± 0.11 | 0.56 ± 0.22 | 0.85 ± 0.12 | **0.65 ± 0.21** | **0.88 ± 0.07** | 0.65 ± 0.12 |
| | OMIGA | 0.97 ± 0.03 | 0.94 ± 0.02 | 0.85 ± 0.02 | 0.95 ± 0.03 | 0.85 ± 0.03 | **0.96 ± 0.03** | 0.86 ± 0.04 | 0.95 ± 0.02 | **0.87 ± 0.02** |
| | CFCQL | 0.90 ± 0.04 | 0.78 ± 0.09 | 0.50 ± 0.11 | 0.70 ± 0.11 | 0.46 ± 0.10 | 0.79 ± 0.07 | 0.53 ± 0.09 | **0.80 ± 0.05** | **0.63 ± 0.08** |
| Average | | 0.94 | 0.85 | 0.66 | 0.82 | 0.62 | 0.86 | 0.68 | **0.87** | **0.71** |
| terran_5_vs_5 | BC | 0.55 ± 0.08 | 0.46 ± 0.02 | 0.38 ± 0.27 | 0.46 ± 0.05 | 0.41 ± 0.13 | 0.51 ± 0.03 | 0.42 ± 0.08 | **0.54 ± 0.08** | **0.46 ± 0.09** |
| | OMIGA | 0.59 ± 0.06 | 0.55 ± 0.15 | 0.54 ± 0.11 | 0.58 ± 0.08 | 0.56 ± 0.04 | 0.61 ± 0.06 | 0.57 ± 0.04 | **0.70 ± 0.06** | **0.61 ± 0.08** |
| | CFCQL | 0.65 ± 0.09 | 0.54 ± 0.12 | 0.49 ± 0.09 | 0.54 ± 0.10 | 0.49 ± 0.19 | 0.53 ± 0.06 | 0.50 ± 0.07 | **0.55 ± 0.11** | **0.52 ± 0.06** |
| zerg_5_vs_5 | BC | 0.32 ± 0.16 | 0.30 ± 0.19 | 0.27 ± 0.13 | 0.28 ± 0.08 | 0.29 ± 0.13 | 0.36 ± 0.07 | 0.33 ± 0.08 | **0.40 ± 0.06** | **0.38 ± 0.06** |
| | OMIGA | 0.44 ± 0.09 | 0.34 ± 0.05 | 0.31 ± 0.09 | 0.36 ± 0.11 | 0.34 ± 0.07 | 0.36 ± 0.13 | 0.33 ± 0.11 | **0.38 ± 0.07** | **0.37 ± 0.13** |
| | CFCQL | 0.55 ± 0.07 | 0.49 ± 0.11 | 0.25 ± 0.09 | 0.47 ± 0.14 | 0.31 ± 0.09 | 0.46 ± 0.07 | 0.34 ± 0.10 | **0.50 ± 0.08** | **0.42 ± 0.15** |
| Average | | 0.51 | 0.44 | 0.37 | 0.45 | 0.40 | 0.47 | 0.42 | **0.51** | **0.46** |
| 4ant | BC | 39.43 ± 6.09 | 25.31 ± 8.15 | 10.48 ± 12.58 | 19.75 ± 3.68 | 10.58 ± 3.38 | 28.30 ± 8.60 | 20.29 ± 6.57 | **31.66 ± 5.10** | **21.89 ± 6.23** |
| | OMIGA | 52.30 ± 5.71 | 33.12 ± 10.60 | 31.76 ± 10.99 | 40.00 ± 8.72 | 30.61 ± 8.46 | 40.50 ± 6.74 | 34.35 ± 4.47 | **48.92 ± 6.57** | **43.09 ± 8.15** |
| | CFCQL | 58.87 ± 6.94 | 38.78 ± 7.85 | 7.10 ± 9.85 | 32.41 ± 5.93 | 6.37 ± 5.79 | 41.98 ± 7.12 | 24.03 ± 8.07 | **48.43 ± 5.91** | **35.28 ± 6.86** |
| Average | | 50.20 | 32.40 | 16.44 | 30.72 | 15.85 | 36.92 | 26.22 | **43.00** | **33.42** |

# H  ADDITIONAL EXPERIMENT RESULTS

## H.1  EVALUATION ON ADDITIONAL ENVIRONMENTS

In this section, we analyze MADiTS's effectiveness on augmenting offline MARL datasets of more tasks, including SMAC 12m, SMACv2 terran_5_vs_5, zerg_5_vs_5, and MAMuJoCo 4ant. The performance of various learned policies on these datasets are presented in Table 4. From the results, we can observe that MA-MBTS demonstrates only slight improvement over Original. MADiff improves upon the Original, highlighting the need for diffusion in data modeling, but overlooks dynamics consistency and data imbalance, leading to a performance gap with MADiTS on imbalanced datasets. On the contrary, MADiTS outperforms the baseline method MA-MBTS and achieves better performance compared to MADiff on most of datasets, highlighting its broad applicability. We can find that MADiTS demonstrates enhanced sample efficiency across various methods, even as the number of agents increases.

## H.2  EVALUATION ON BALANCED OFFLINE DATASETS

To thoroughly evaluate the general effectiveness of MADiTS, we also utilize the multi-agent offline dataset in MPE with continuous action space built in OMAR (Pan et al., 2022) for data augmentation, testing the effectiveness of our method on balanced datasets. This dataset comprises four different quality datasets collected from policies of varying quality trained by MATD3 (Ackermann et al., 2019), including Expert, Medium, Medium-Replay, and Random. The Expert, Medium and Random datasets consist of one million samples each, generated by fully trained policies, medium-performing policies, or a random policy in an online environment. The Medium-Replay dataset records all data in the replay buffer before the policy reaching medium performance level. We assess the enhancement in dataset quality using behavior cloning policy, measuring the average return of the BC policy in the online environment as the quality metric.

From the results shown in Table 5, we can first observe that MA-MBTS demonstrates only slight improvement over Original, while MADiTS achieves the best performance despite the varying quality of datasets, which proves the robustness and generality of MADiTS. Notably, in the mixed-

Table 5: Evaluation results of multi-agent balanced datasets. The mean and standard error are computed based on the average return of BC policies trained on the datasets, with 5 random seeds.

| Envs | Dataset | Original | MA-MBTS | MADiTS |
|---|---|---|---|---|
| CN | expert | 100.42 ± 5.67 | 101.70 ± 3.13 | **127.91 ± 5.63** |
| | medium | 80.46 ± 5.05 | 81.20 ± 8.21 | **91.31 ± 2.24** |
| | md-replay | 32.17 ± 4.51 | 55.53 ± 27.10 | **84.47 ± 8.04** |
| | random | 2.69 ± 4.24 | 2.79 ± 4.43 | **22.38 ± 4.59** |
| PP | expert | 77.54 ± 5.73 | 79.47 ± 9.96 | **97.04 ± 11.81** |
| | medium | 54.14 ± 10.19 | 53.19 ± 7.99 | **63.25 ± 10.15** |
| | md-replay | -6.43 ± 3.21 | -6.72 ± 2.39 | **25.04 ± 11.45** |
| | random | -8.94 ± 1.55 | -9.12 ± 1.36 | **-7.38 ± 2.49** |
| World | expert | 76.89 ± 23.08 | 73.26 ± 17.45 | **138.10 ± 34.01** |
| | medium | 54.99 ± 14.27 | 62.62 ± 14.99 | **96.27 ± 11.78** |
| | md-replay | 19.28 ± 4.06 | 20.49 ± 2.09 | **29.46 ± 4.65** |
| | random | 6.19 ± 3.08 | 5.92 ± 2.95 | **14.01 ± 2.95** |

quality md-replay datasets, the enhanced datasets after stitching can rival the quality of the medium dataset. This highlights that MADiTS effectively leverages a small amount of high-quality data to improve the quality of other suboptimal data.

### H.3 FULL PARAMETER STUDIES

Apart from the sensitivity analysis of the reconstruction threshold $\delta_{\text{recon}}$ in Section 5.4, we also test the sensitivity of two key hyperparameters, augmentation ratio $r_{\text{aug}}$ and regeneration limit $l_{\text{limit}}$.

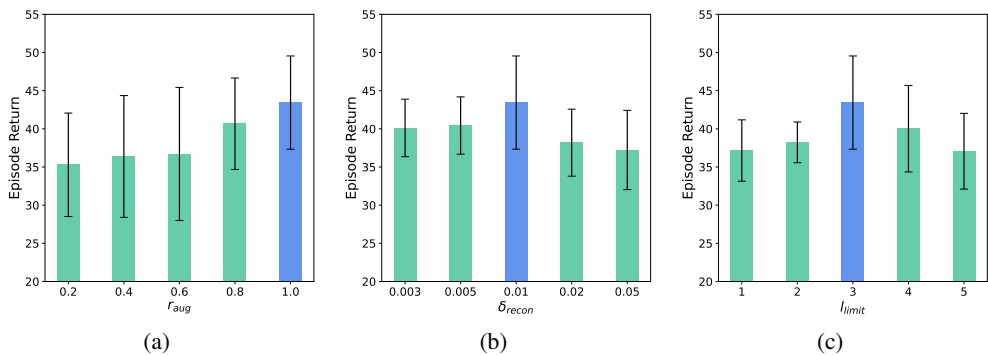

(a)                               (b)                               (c)

Figure 5: Average returns on exp-m dataset in Cooperative Navigation of BC policies as the hyperparameters change. The errors bars represent the standard error of 5 different random seeds.

From the results shown in Figure 5, we observe that as $r_{\text{aug}}$ gradually decreases, the performance of the new dataset slowly declines; however, it still shows considerable improvement compared to the original dataset. This indicates that the trajectories generated through stitching can overcome the imbalances in the original dataset, with only a small portion effectively enhancing the dataset's quality. Furthermore, the impact of $l_{\text{limit}}$ remains the same as $\delta_{\text{recon}}$. As $l_{\text{limit}}$ increase, the value first rise and then fall, which also reflects the negative impact of either overly lenient or overly strict constraint like what we observe on $\delta_{\text{recon}}$.

### H.4 EVALUATION ON MORE ADDITIONAL PERTURBATION SETTINGS

Besides the experiments conducted on datasets with different timesteps of perturbations $t_{\text{imb}}$, we also explore the impact of qualities of behavior policy to the augmentation effects by applying

Table 6: Evaluation results on more additional perturbation settings on Cooperative Navigation task. The mean and standard error are computed based on the average return of BC policies trained on the datasets, with 5 different random seeds.

| Dataset | $t_{\mathrm{imb}} = 2$ | | $t_{\mathrm{imb}} = 4$ | | $t_{\mathrm{imb}} = 6$ | | $t_{\mathrm{imb}} = 8$ | |
|---------|------|------|------|------|------|------|------|------|
| | before | after | before | after | before | after | before | after |
| md-m | -41.81 ± 3.42 | **-34.58 ± 4.18** | -43.17 ± 3.13 | **-37.40 ± 3.02** | -42.14 ± 3.76 | **-36.12 ± 2.84** | -42.39 ± 2.74 | **-36.08 ± 3.99** |
| md-s | -48.64 ± 1.41 | **-40.88 ± 2.67** | -51.40 ± 2.95 | **-42.28 ± 4.22** | -50.57 ± 1.12 | **-40.04 ± 2.76** | -51.98 ± 3.27 | **-42.50 ± 2.92** |

perturbations to medium-performing policies and collect more imbalanced datasets, denoted as md-m and md-s. The average returns of the BC policy on these datasets are shown in Table 6, from which we can observe that the trend of enhancement is consistent with the results of that in imbalanced datasets collect by expert-performing behavior policies.

## H.5 ANALYSIS OF OFFLINE CREDIT ASSIGNMENT ACCURACY BY INTEGRATED GRADIENT

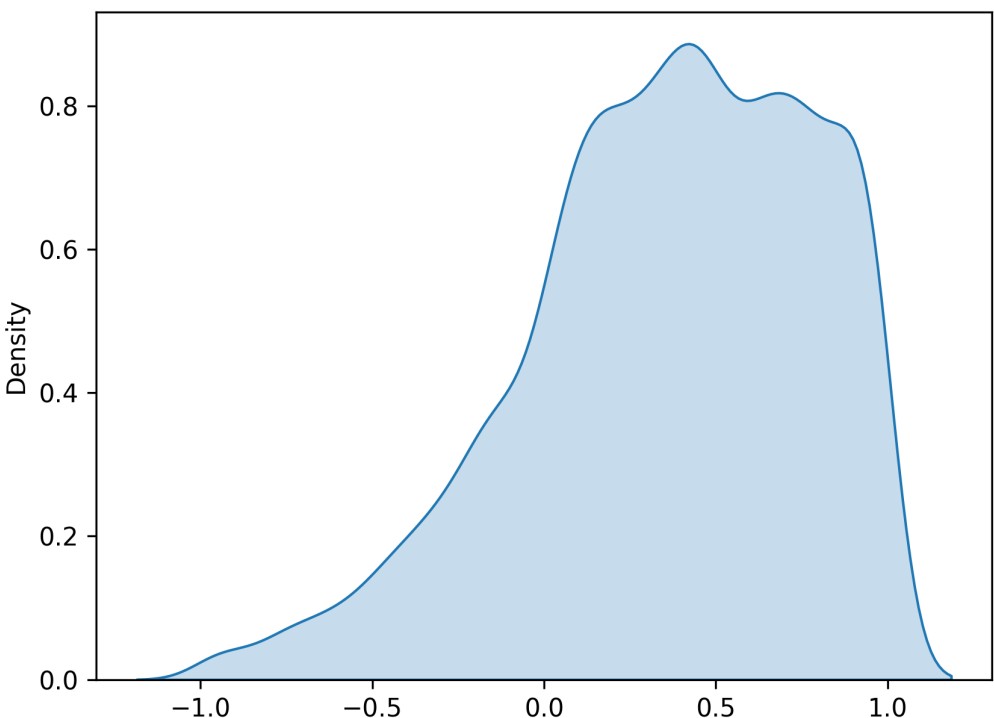

Figure 6: Distribution visualization by kernel density estimation of the average Kendall correlation coefficients for trajectory segments compared with true data in the exp-m dataset of the CN task. A coefficient less than 0 indicates negative correlation, greater than 0 indicates positive correlation, and 0 indicates independence.

In this section, we analyze the accuracy of the credit assignment method in our method. In Cooperative Navigation, the closer a landmark is to an agent, the higher the return the team can achieve. Based on our understanding of the environment's true reward function, we can derive each agent's actual contribution at every time step as ground truth (this oracle information is not used in our method and is only for analysis). Therefore, we calculate the true individual contributions of agents from the offline dataset $\mathcal{D}$ and rank the contributions for each agent at every time step as $\mathrm{rank}^i_{t,\mathrm{oracle}}$.

The similarity between the true ranking and the estimated ranking is measured using the Kendall correlation coefficient (Kendall, 1945). This coefficient evaluates the monotonic relationship between two ordinal variables, ranging from $[-1, 1]$. A coefficient less than 0 indicates a negative correlation, greater than 0 indicates a positive correlation, and a value of 0 implies independence. The closer

the absolute value of the coefficient is to 1, the stronger the relationship, making it well-suited for assessing the consistency between the estimated and true rankings. The mean Kendall correlation coefficient for each trajectory segment is computed as:

$$\tau_{\text{kendall}}^{\text{mean}}(\boldsymbol{\tau}) = \frac{1}{T-t} \sum_{t'=t}^{T-1} \tau_{\text{kendall}}\left(\left(\text{rank}_{t'}^1, \cdots, \text{rank}_{t'}^n\right), \left(\text{rank}_{t',\text{oracle}}^1, \cdots, \text{rank}_{t',\text{oracle}}^n\right)\right),$$

where $\tau_{\text{kendall}}$ is directly implemented using the 'scipy' library in Python3. Considering that offline credit assignment is performed on high-return trajectory segments generated by the diffusion model, we apply integrated gradient (IG) to compute the contribution of each agent for the trajectory segment dataset in the Cooperative Navigation exp-m dataset. We then calculate the average Kendall correlation coefficient for each trajectory segment and plotted its distribution as a histogram.

The distribution of the computed Kendall correlation coefficients is presented in Figure 6. It can be observed that the majority of trajectory segments have an average Kendall correlation coefficient greater than 0, demonstrating that the offline credit assignment method based on Integrated Gradients achieves a high level of accuracy in practice.

### H.6 Diversity Visualization of Synthesized Trajectories

Some previous works (Tian et al., 2023; Li et al., 2023) have emphasized the importance of diverse trajectories in offline multi-agent reinforcement learning. Here, we examine how our data augmentation method impacts the diversity of trajectories in the augmented dataset. Figure 7 visualizes 8 generated trajectories with identical agents starting positions of and landmarks positions in the augmented CN exp-m dataset, comparing results from w/o DC+PN and MADiTS, respectively.

As observed in Figure 7(a), the inherent diversity of the diffusion models (Ho et al., 2020) allows w/o DC + PN to exhibit considerable diversity, despite issues of dynamics consistency and suboptimal performance. In contrast, after incorporating our dynamics constraints and behavior correction, each agent's paths to the landmarks still show significant diversity, indicating the capabilities of MADiTS to maintain impressive diversity while simultaneously enhancing trajectory quality.

## I Further Discussions on MADiTS

### I.1 Application to Cooperative-competitive Settings

Our paper mainly addresses the cooperative multi-agent reinforcement learning (MARL) problem, a widely studied setup where all agents share a global reward. Through extensive experiments across various offline MARL environments, we demonstrate that MADiTS significantly improves MARL performance. While our current focus is on cooperative settings, the method can be naturally extended to cooperative-competitive scenarios by equipping each team with its own MADiTS model. In this setup, teams can independently learn model parameters using tailored buffers for training. We leave the further exploration of this topic to future work.

### I.2 Concerns on the Use of Diffusion Models

Biases in the original offline dataset can propagate into the diffusion model, potentially affecting the quality of the generated trajectories. To mitigate this, our method employs a bidirectional dynamics constraint, ensuring that the generated trajectories remain consistent with the environmental dynamics. Additionally, we integrate an offline credit assignment technique to identify and optimize the performance of underperforming agents within the generated trajectory segments, further enhancing the overall quality and utility of the augmented data.

On one hand, generative models like ChatGPT (Achiam et al., 2023) and SORA (OpenAI, 2024) rely on large-scale datasets to train architectures such as Transformers or diffusion-based models. These models exhibit exceptional generative capabilities across domains like language and video, aligning with scaling laws that link performance to data size. Recognizing the importance of data, these methods often use autoregressive training or advanced techniques to optimize data fitting. For real-world applications such as autonomous driving (Wang et al., 2024b), healthcare (Kazerouni et al.,

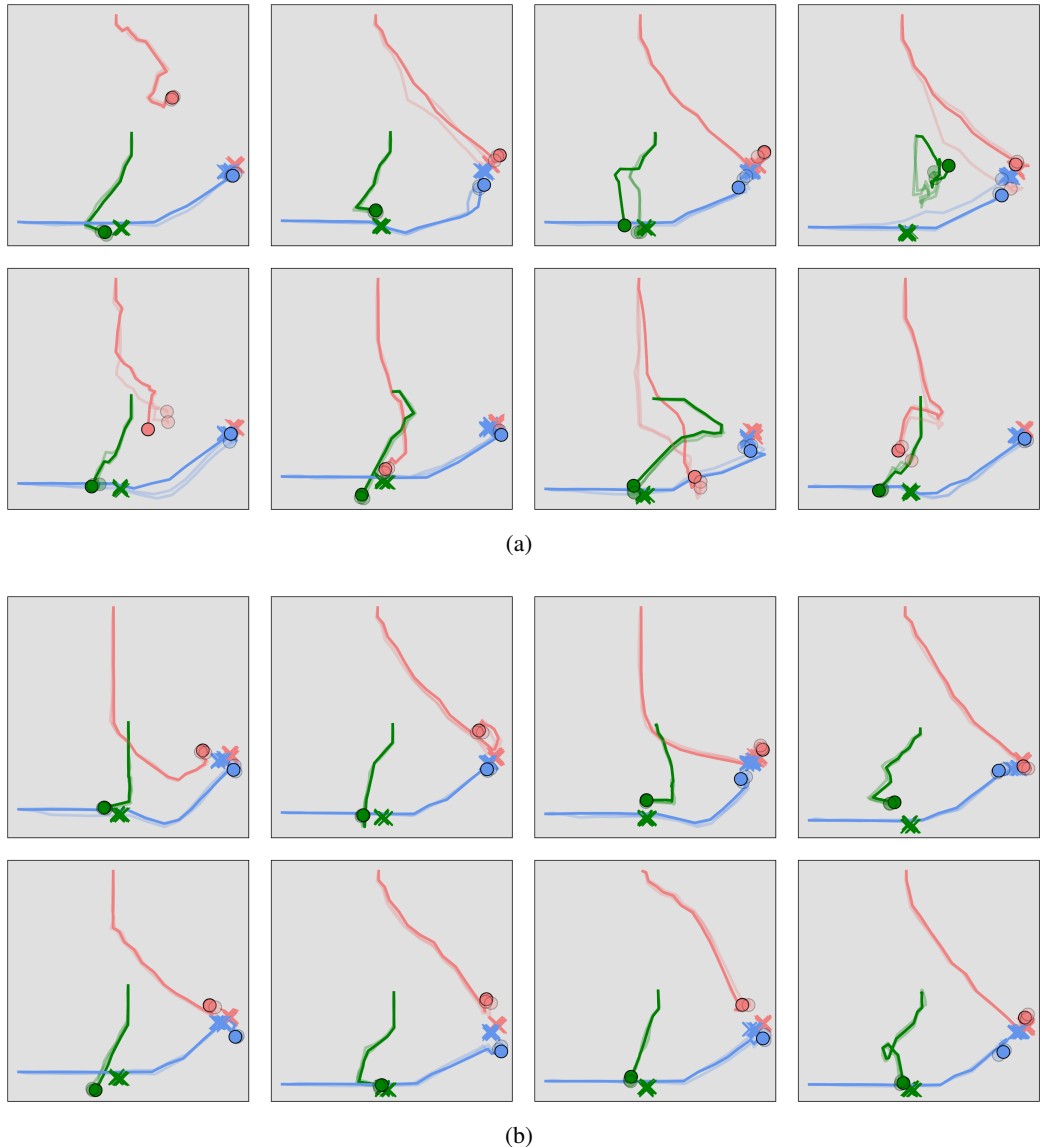

Figure 7: Diversity visualization of 8 generated trajectories with identical starting positions of agents and positions of landmarks in the augmented CN exp-m dataset. Lighter-colored trajectories are the ones extracted from observations of other agents. (a) Trajectories generated by w/o DC+PN. (b) Trajectories generated by MADiTS.

2023), and finance (Wang & Ventre, 2024), diffusion models require extensive and diverse datasets to ensure robust performance. Recent advancements have highlighted the potential of diffusion models to transform these domains.

On the other hand, despite their capabilities, these methods face challenges such as unreliable or unrepresentative data. To ensure reliability in real-world applications, techniques like human-in-the-loop testing (Singi et al., 2024) and risk control mechanisms (Yang et al., 2021a) are crucial.

To address the issue of synthetic data deviating from real-world distributions, our method MADiTS introduces a bidirectional dynamics constraint to align generated trajectories with environmental dynamics. Moreover, the offline credit assignment technique enhances robustness by identifying and improving underperforming agents in generated segments. Experimental results validate the effectiveness of MADiTS in overcoming these challenges.

