# OpenReview forum: "Efficient Multi-agent Offline Coordination via Diffusion-based Trajectory Stitching"
_ICLR.cc/2025/Conference — ICLR 2025 Poster_

### Official Review · Reviewer_bitr · 2024-10-30

**Soundness:** 2
**Presentation:** 3
**Contribution:** 2
**Rating:** 6
**Confidence:** 4

**Summary:**

The authors propose a method called MADiTS to enhance the quality of imbalanced offline datasets in Multi-Agent Reinforcement Learning (MARL). By applying diffusion-based data augmentation, MADiTS generates high-return trajectories while enforcing dynamics constraints to ensure they align with the environment dynamics. Additionally, a behavior correction mechanism correct the actions of suboptimal agents, improving their trajectories within shared reward contexts. Experiments on MPE and SMAC datasets show that MADiTS can improve the performance of offline algorithms like Behavior Cloning (BC) and other offline algorithms.

**Strengths:**

- The authors address an important issue of dataset quality in MARL, particularly relevant for offline applications where data imbalance often impacts policy performance in real-world scenarios.
- The paper is well-written, clear and easy to follow.
- The authors apply their method to datasets collected on SMAC and MPE, two popular MARL environments, and apply it to behavior cloning (BC) and two modern offline MARL algorithms - OMIGA and CFCQL.
- The authors provide an ablation study to show the impact of different components of their MADiTS method.

**Weaknesses:**

Methodology:
- The authors provide details on how they generate the imbalanced dataset, using random perturbations on expert trajectories. However, it is not clear why this is better than using a dataset collected by mixed policies, e.g. taking medium and expert policies and combining their trajectories in a dataset. This appears to be more realistic and relevant to the problem of imbalanced datasets in MARL. A discussion on this choice would be relevant.
- Work such as [2,3] highlights the importance of diverse trajectories in offline MARL. This impact of MADiTS on the diversity of the trajectories is not discussed in the paper.
- Clarity is needed regarding the computational cost of MADiTS. Details such as runtime and comparison with other trajectory augmentation methods (e.g., MA-MBTS or baseline models) would be relevant, especially since the text includes vague descriptors like “until a satisfactory trajectory is obtained.” Further, insights into how MADiTS scales with agent numbers or in larger MARL environments would enhance applicability.

Comparisons with Related Diffusion Models:
- In the appendix, the authors mention they use MaDiff [1] architecture for the diffusion model. However, the original MaDiff model is not included as a baseline in the main experiments, nor is Diffusion Offline Multi-agent Model (DOM2) [3], another relevant diffusion-based MARL trajectory augmentation approach. Furthermore, Shared Individual Trajectories (SIT) [2], another work that aims at helping with imbalanced datasets in offline MARL is only briefly mentioned in the appendix, and not mentioned in the main text.  A comparison with MaDiff (or clarification if “w/o dc + pn” in Figure 4a is equivalent to MaDiff) would strengthen the evaluation. Including DOM2 and SIT as baselines or discussing the differences would clarify MADiTS's unique contributions to diffusion-based MARL augmentation.

1] Zhu Z, Liu M, Mao L, Kang B, Xu M, Yu Y, Ermon S, Zhang W. Madiff: Offline multi-agent learning with diffusion models. arXiv preprint arXiv:2305.17330. 2023 May 27.

2] Tian Q, Kuang K, Liu F, Wang B. Learning from good trajectories in offline multi-agent reinforcement learning. In Proceedings of the AAAI Conference on Artificial Intelligence 2023 Jun 26 (Vol. 37, No. 10, pp. 11672-11680).

3] Li Z, Pan L, Huang L. Beyond conservatism: Diffusion policies in offline multi-agent reinforcement learning. arXiv preprint arXiv:2307.01472. 2023 Jul 4.

**Questions:**

1. Why does this approach generate an imbalanced dataset by perturbing expert trajectories? Why is this better than using a dataset collected by mixed policies and combine their trajectories in a dataset?
2. How does the proposed method impact the diversity of the trajectories in the augmented dataset? Do the dynamics constraints or behavior correction mechanism hurt the diversity of the trajectories?
3. What is the computational cost of the proposed method compared to the baselines? How does the proposed method scale with the number of agents, particularly with the integrated gradient computation?
4. How does MADiTS compare to standalone MaDiff, DOM2 and SIT? What are the key differences, and why were these methods not used as baselines in the primary experiments?

---

> ### Author Response · Authors · 2024-11-20
> **Response to Reviewer bitr (Part I)**
>
> Thank you very much for carefully reviewing our paper and providing constructive comments and suggestions. In response, we have added experiments and comparisons with different methods, which are presented as follows:
>
> **Q1: Reasons of generating the imbalanced dataset using random perturbations on expert trajectories.**
>
> A1: Our primary goal in generating imbalanced datasets is to create scenarios more challenging than those in mixed-quality datasets by explicitly introducing temporal and spatial imbalances. This approach pushes the boundaries of data augmentation techniques in Offline MARL. To achieve this, we apply random perturbations to expert trajectories, simulating both types of imbalances. This method constructs datasets with more pronounced imbalances compared to simple mixing strategies, allowing for a more rigorous evaluation of our method under challenging task settings.
>
> Building on a general framework for addressing data imbalance in data augmentation [1], our approach extends beyond spatial imbalance to also tackle temporal imbalance. This expansion broadens the scope of our method and introduces additional challenges, enabling exploration of more diverse and realistic multi-agent settings. These innovations highlight the unique aspects of our approach.
>
> Regarding to the performance of our method on the mixed-quality datasets mentioned by the reviewer, as shown in the simplified table below, MADiTS still significantly enhances dataset quality in mixed-quality md-replay datasets (with lines bolded). In tested environments, its performance approaches or even matches that of medium-quality datasets, demonstrating its ability to effectively leverage limited high-quality data to improve suboptimal data segments. For more detailed results, please refer to Table 5 in the revised manuscript.
>
> | Env | Dataset | Original | MADiTS |
> | --- | --- | --- | --- |
> | CN | expert | 100.42 ± 5.67 | 127.91 ± 5.63 |
> |  | medium | 80.46 ± 5.05 | 91.31 ± 2.24 |
> |  | **md-replay** | **32.17 ± 4.51** | **84.47 ± 8.04** |
> |  | random | 2.69 ± 4.24 | 22.38 ± 4.59 |
> | PP | expert | 77.54 ± 5.73 | 97.04 ± 11.81 |
> |  | medium | 54.14 ± 10.19 | 63.25 ± 10.15 |
> |  | **md-replay** | **-6.43 ± 3.21** | **25.04 ± 11.45** |
> |  | random | -8.94 ± 1.55 | -7.38 ± 2.49 |
> | World | expert | 76.89 ± 23.08 | 138.10 ± 34.01 |
> |  | medium | 54.99 ± 14.27 | 96.27 ± 11.78 |
> |  | **md-replay** | **19.28 ± 4.06** | **29.46 ± 4.65** |
> |  | random | 6.19 ± 3.08 | 14.01 ± 2.95 |
>
> **Q2: The impact of MADiTS on the diversity of the trajectories.**
>
> A2: The dynamics constraints and behavior correction mechanism not only preserve the diversity of trajectories but also significantly improve data quality. To illustrate this, we provide visualizations of the synthesized trajectories under a fixed, non-stochastic environment's initial state, defined by the agents' starting positions and landmark locations. Comparisons between MADiTS and its variant without dynamics constraints or behavior correction reveal that our approach generates data with broader coverage and higher diversity. Further details and analysis can be found in Figure 7 and Appendix H.6.
>
> **Q3: Clarity regarding the computational cost of MADiTS and vague descriptors.**
>
> A3: Our method achieves computational complexity comparable to other baseline approaches. To demonstrate this, we provide a summary of the computational costs for MADiTS and baseline methods on the Cooperative Navigation task. The experiments were conducted on servers equipped with GeForce RTX 2080 Ti GPUs. As shown in the table below, MADiTS maintains computational complexity similar to other methods, staying within an acceptable range while ensuring competitive performance.
>
> | **Method** | **Model Training (GPU Hours)** | **Trajectory Processing (GPU Hours)** | **Total (GPU Hours)** |
> | --- | --- | --- | --- |
> | **MADiTS** | 36 | 4 | 40 |
> | **MADiff** | 36 | - | 36 |
> | **MA-MBTS** | - | 48 | 48 |
>
> As for the vague descriptors, we have changed the sentence from “This iterative procedure continues until we obtain a satisfactory augmented episode trajectory” to “This iterative procedure continues until a satisfactory augmented episode trajectory is generated **within the predefined limit or is discarded otherwise**”.  We greatly appreciate your suggestion.

---

> > ### Author Response · Authors · 2024-11-20
> > **Response to Reviewer bitr (Part II)**
> >
> > **Q4: How MADiTS performs in larger MARL environments.**
> >
> > A1: MADiTS enhances sample efficiency across various methods, even as the number of agents increases. To validate its effectiveness, we conducted experiments on the 12m map from SMAC, with 12 marine agents and 12 marine enemies. As shown in the table below, MADiTS outperforms the baseline method MA-MBTS and shows competitive results compared to MADiff, demonstrating its broad applicability. As for the large scale MARL setting with hundreds or thousands of agents[2], solving the scalability issue by techniques like grouping would be of great interest, and we leave it in the future work.
> >
> > | Envs | Algs | Balanced | Original |  | MA-MBTS |  | MADiff |  | MADiTS |  |
> > | --- | --- | --- | --- | --- | --- | --- | --- | --- | --- | --- |
> > |  |  | exp | exp-m | exp-s | exp-m | exp-s | exp-m | exp-s | exp-m | exp-s |
> > | 12m | BC | 0.95 ± 0.03 | 0.83 ± 0.11 | 0.63 ± 0.28 | 0.81 ± 0.11 | 0.56 ± 0.22 | 0.85 ± 0.12 | **0.65 ± 0.21** | **0.88 ± 0.07** | **0.65 ± 0.12** |
> > |  | OMIGA | 0.97 ± 0.03 | 0.94 ± 0.02 | 0.85 ± 0.02 | 0.95 ± 0.03 | 0.85 ± 0.03 | **0.96 ± 0.03** | 0.86 ± 0.04 | 0.95 ± 0.02 | **0.87 ± 0.02** |
> > |  | CFCQL | 0.90 ± 0.04 | 0.78 ± 0.09 | 0.50 ± 0.11 | 0.70 ± 0.11 | 0.46 ± 0.10 | 0.79 ± 0.07 | 0.53 ± 0.09 | **0.80 ± 0.05**  | **0.63 ± 0.08** |
> > | Average |  | 0.94 | 0.85 | 0.66 | 0.82 | 0.62 | 0.86 | 0.68 | **0.87** | **0.71** |
> >
> > **Q5: Comparison with MADiff, DOM2, and SIT.**
> >
> > A5: Our method focuses on addressing sample efficiency issues by leveraging diffusion models to perform trajectory stitching for data augmentation in temporally and spatially imbalanced datasets, allowing offline MARL algorithms to achieve better performance by learning from the enhanced dataset, and is different from the mentioned works:
> >
> > - MADiff[3] is a diffusion model-based offline MARL algorithm designed to predict future joint actions for decision-making by modeling teammate behaviors. It uses a diffusion model to capture joint observation sequences and infer actions for planning, achieving strong performance on balanced datasets and excelling in standard offline MARL settings. In contrast, MADiTS specifically targets the challenges of data imbalance in offline MARL, focusing on improving sample efficiency through innovative data augmentation techniques. While MADiff enhances learning efficiency in balanced scenarios, MADiTS addresses the unique difficulties posed by imbalanced datasets. To highlight these distinctions, we compare the performance of MADiff, extended for data augmentation, with MADiTS across various environments. The results demonstrate MADiTS’s superior effectiveness in tackling data imbalance challenges.
> >
> > | Envs | Algs | Original |  | MADiff |  | MADiTS |  |
> > | --- | --- | --- | --- | --- | --- | --- | --- |
> > |  |  | exp-m | exp-s | exp-m | exp-s | exp-m | exp-s |
> > | CN | BC | 17.27 ± 3.66 | 9.03 ± 3.40 | 40.44 ± 3.75 | 30.24 ± 4.82 | **43.44 ± 6.11** | **37.82 ± 4.45** |
> > |  | OMIGA | -2.27 ± 57.07 | -11.60 ± 58.90 | 19.26 ± 67.68 | 7.27 ± 66.89 | **23.02 ± 69.69** | **22.91 ± 44.94** |
> > |  | CFCQL | -33.40 ± 44.28 | -56.44 ± 40.38 | 30.08 ± 12.97 | 21.88 ± 10.71 | **39.57 ± 16.14** | **28.60 ± 20.94** |
> > | PP | BC | 48.49 ± 4.69 | 49.21 ± 3.90 | 49.55 ± 6.50 | 49.93 ± 3.74 | **54.85 ± 4.23** | **55.50 ± 4.28** |
> > |  | OMIGA | 37.90 ± 25.34 | 26.73 ± 40.67 | 47.48 ± 20.97 | 58.44 ± 4.76 | **63.02 ± 3.40** | **63.71 ± 5.67** |
> > |  | CFCQL | 45.03 ± 4.62 | 28.88 ± 6.29 | 45.50 ± 6.76 | 30.04 ± 7.02 | **47.38 ± 3.74** | **32.25 ± 10.98** |
> > | World | BC | 47.29 ± 3.00 | 48.30 ± 5.96 | 50.79 ± 2.62 | 51.55 ± 4.96 | **54.25 ± 4.35** | **52.84 ± 5.29** |
> > |  | OMIGA | 54.23 ± 6.20 | 52.92 ± 5.64 | 49.55 ± 24.99 | 48.26 ± 24.75 | **57.35 ± 5.89** | **58.59 ± 9.32** |
> > |  | CFCQL | 28.59 ± 1.90 | 18.14 ± 18.16 | 28.63 ± 10.14 | 34.35 ± 6.91 | **29.62 ± 10.91** | **39.36 ± 1.91** |
> > | Average |  | 27.01 | 18.35 | 40.14 | 36.88 | **45.83** | **43.50** |
> > | 2m_vs_1z | BC | 0.03 ± 0.06 | 0.03 ± 0.06 | 0.30 ± 0.18 | 0.30 ± 0.18 | **0.35 ± 0.20** | **0.35 ± 0.20** |
> > |  | OMIGA | 0.59 ± 0.28 | 0.59 ± 0.28 | 0.97 ± 0.04 | 0.97 ± 0.04 | **0.98 ± 0.02** | **0.98 ± 0.02** |
> > |  | CFCQL | 0.72 ± 0.24 | 0.72 ± 0.24 | 0.92 ± 0.05 | 0.92 ± 0.05 | **0.94 ± 0.05** | **0.94 ± 0.05** |
> > | 3m | BC | 0.42 ± 0.05 | 0.38 ± 0.20 | 0.44 ± 0.18 | 0.36 ± 0.15 | **0.51 ± 0.19** | **0.43 ± 0.18** |
> > |  | OMIGA | 0.93 ± 0.05 | 0.90 ± 0.03 | **1.00 ± 0.00** | 0.94 ± 0.03 | **1.00 ± 0.00** | **0.96 ± 0.03** |
> > |  | CFCQL | 0.90 ± 0.06 | 0.80 ± 0.08 | 0.89 ± 0.05 | 0.81 ± 0.09 | **0.94 ± 0.07** | **0.89 ± 0.06** |
> > | 2s3z | BC | 0.73 ± 0.11 | 0.68 ± 0.14 | 0.75 ± 0.06 | 0.67 ± 0.05 | **0.78 ± 0.06** | **0.69 ± 0.26** |
> > |  | OMIGA | 0.86 ± 0.16 | 0.67 ± 0.05 | **1.00 ± 0.00** | 0.71 ± 0.15 | **1.00 ± 0.00** | **0.76 ± 0.19** |
> > |  | CFCQL | 0.73 ± 0.15 | 0.66 ± 0.14 | 0.80 ± 0.09 | 0.63 ± 0.15 | **0.92 ± 0.02** | **0.68 ± 0.26** |
> > | Average |  | 0.65 | 0.60 | 0.78 | 0.70 | **0.82** | **0.74** |
> >
> > (to be continued)

---

> > > ### Author Response · Authors · 2024-11-20
> > > **Response to Reviewer bitr (Part III)**
> > >
> > > **Q5: Comparison with MADiff, DOM2, and SIT.**
> > >
> > > A5: (continued)
> > > - SIT[1] is an offline MARL method that focuses on learning an effective joint policy from agent-wise imbalanced datasets, primarily addressing spatial imbalance. It achieves this through an attention-based reward decomposition network for offline credit assignment, which identifies high-quality individual trajectories for sharing among agents, and a graph attention network for conservative policy training. In contrast, our method, MADiTS, prioritizes improving sample efficiency through a data-oriented augmentation pipeline. By employing a diffusion model-based trajectory stitching mechanism, MADiTS enhances dataset quality to address both temporal and spatial imbalances. Unlike SIT, which directly learns policies from imbalanced datasets, MADiTS generates augmented datasets that can be flexibly used by any offline MARL algorithm, providing enhanced flexibility and modularity.
> > > - DOM2[4] is a diffusion model-based offline MARL algorithm that aims to enhance policy expressiveness and diversity. It achieves notable improvements in performance, generalization, and data efficiency through an accelerated solver for diffusion-based policy construction and a policy regularizer, while also scaling up dataset size to boost policy learning. DOM2 excels on balanced datasets, outperforming traditional conservatism-based offline MARL methods. In contrast, MADiTS targets the challenges posed by imbalanced datasets. By addressing temporal and spatial imbalances, MADiTS improves dataset quality, enabling other offline MARL algorithms to achieve better performance and demonstrating its effectiveness in handling imbalanced scenarios.
> > >
> > > Thanks for your constructive suggestions. We have included MADiff[3] in the main experiments as a baseline. And we also discuss the key differences with these related mentioned methods in detail in Appendix B and more related works in the revised manuscript.
> > >
> > > **References:**
> > >
> > > [1] Tian, Q., et al. Learning from good trajectories in offline multi-agent reinforcement learning. In *Association for the Advancement of Artificial Intelligence*, pp. 11672–11680, 2023.
> > >
> > > [2] Zheng, L., et al. Magent: A many-agent reinforcement learning platform for artificial collective intelligence. In *Association for the Advancement of Artificial Intelligence*. 2018, 32(1).
> > >
> > > [3] Zhu, Z., et al. MADiff: Offline multi-agent learning with diffusion models. *arXiv preprint arXiv:2305.17330*, 2023.
> > >
> > > [4] Li, Z., et al. Beyond conservatism: Diffusion policies in offline multi-agent reinforcement learning. *arXiv preprint arXiv:2307.01472*, 2023.

---

> ### Author Response · Authors · 2024-11-24
> **Dear Reviewer bitr, are our responses address your questions?**
>
> Dear Reviewer bitr:
>
> We thank you again for your comments and hope our responses could address your questions. As the response system will end in four days, please let us know if we missed anything. More questions on our paper are always welcomed.
>
> Sincerely yours,
>
> Authors of Paper10913

---

> ### Comment · Reviewer_bitr · 2024-11-25
> **Follow Up**
>
> Thanks to the authors for their detailed response and for addressing some of my initial questions. I have two follow-up queries for further clarification:
>
> 1. What was the computational cost of the 12m experiments across the different methods (MADiTS, MADiff, MA-MBTS)? This would help better understand how the approach scales computationally with respect to the number of agents.
> 2. Do you have quantifiable measures of agent diversity? While examining trajectories is informative, it may not adequately capture or quantify diverse behaviours in a rigorous way.

---

> > ### Author Response · Authors · 2024-11-25
> > **Thank you for your continued attention to our work!**
> >
> > Thank you for your continued attention to our work. We appreciate your further inquiry, and we are happy to clarify the points raised.
> >
> > **Q1: computational cost of the 12m experiments.**
> >
> > The computational cost of model training for the compared methods increases slightly but remains within an acceptable range. To illustrate, we present the specific computational cost for a single training run on the SMAC task, using the 3m and 12m environments as examples:
> >
> > | **Map** | **Method** | **Model Training (GPU Hours)** | **Trajectory Processing (GPU Hours)** | **Total (GPU Hours)** |
> > | --- | --- | --- | --- | --- |
> > | **3m** | **MADiTS** | 35 | 4 | 39 |
> > |  | **MADiff** | 35 | - | 35 |
> > |  | **MA-MBTS** | - | 43 | 43 |
> > | **12m** | **MADiTS** | 38 | 6 | 44 |
> > |  | **MADiff** | 37 | - | 37 |
> > |  | **MA-MBTS** | - | 54 | 54 |
> >
> > When dealing with environments involving hundreds or thousands of agents, addressing the scalability challenge through techniques such as grouping becomes a topic of great interest. We acknowledge this valuable suggestion and plan to explore it in future work. Thank you for raising this important question.
> >
> > **Q2: quantifiable measures of agent diversity.**
> >
> > Building on the illustration provided in Figure 7, we proceed with a quantitative analysis using the mean Average Displacement Error (mean ADE) [1], a distance-based metric for evaluating the diversity of new trajectories generated from fixed initial states. The mean ADE is calculated as follows:
> >
> > $$
> > \text{mean ADE} = \mathbb{E} _{(\tau_i, \tau_j)} \left[\frac{1}{T}\sum _{t=1}^T \mathscr{D}(p_t^i, p_t^j)\right]
> > $$
> > where $\tau_i$ and $\tau_j$ denote two trajectories sampled from the augmented dataset generated under identical initial states. The positions of the agents at timestep $t$ are represented by $p_t^i$ and $p_t^j$, respectively, and $\mathscr{D}$ is the distance metric, defined here as the Euclidean distance. To estimate the expectation, we fixed identical initial states and generated 256 trajectories using our MADiTS method through trajectory stitching. This metric evaluates the diversity of behaviors, where a higher value indicates greater diversity.
> >
> > We conducted experiments using five different sets of fixed initial states. For MADiTS, the mean ADE is **is 0.184 ± 0.078**, whereas its variant without dynamics constraints or behavior correction shows a mean ADE of **0.213 ± 0.116**. Our method significantly improves overall performance, the reduction in mean ADE is only limited to 13%. It is important to note that under fixed initial states, the theoretical maximum return is unique and inherently lacks diversity. Thus, we consider the relationship between high returns and diversity to be a trade-off. Our method achieves higher performance while maintaining comparable diversity levels.
> >
> > Thank you all again for your time and valuable advice!  Feel free to let us know if you have any more comments or questions.
> >
> >
> > **References:**
> >
> > [1] Pellegrini, S., Ess, A., Schindler, K., Van Gool, L.: You’ll never walk alone: Modeling social behavior for multi-target tracking. In: 2009 IEEE 12th International Conference on Computer Vision. pp. 261–268. IEEE (2009)

---

> > > ### Comment · Reviewer_bitr · 2024-11-26
> > >
> > > Thanks for the detailed response. After considering the rebuttal points, I will increase my score by one point.

---

> > > > ### Author Response · Authors · 2024-11-27
> > > >
> > > > Thank you for your time in our paper, and supporting the community.

---

### Official Review · Reviewer_qWY9 · 2024-10-31

**Soundness:** 3
**Presentation:** 2
**Contribution:** 3
**Rating:** 8
**Confidence:** 4

**Summary:**

The paper presents a novel data augmentation method, MADiTS, to generate high-quality multi-agent trajectories for offline policy learning. A multi-agent trajectory diffusion model is trained to generate augmented future trajectories conditioned on the first-step joint observations. Multiple techniques are proposed to improve the quality of augmented trajectories. First, a forward dynamics model is used together with the inverse dynamics model to filter out dyanmic inconsist segments. Then, the interagted gradient is adopted to determine underperforming agents, and regenerating those agents' trajectories while keeping others' fixed. Experimental results on MPE and SMAC datasets demonstrate superior performence of MADiTS over baseline augmentation methods.

**Strengths:**

1. The paper is well-written and easy to follow.

2. The idea of using the interagted gradient to find underperforming agents is novel to me, and I believe it might have border potential in multi-agent learning.

3. The experiment results are impressive, achieving SOTA performance on all datasets.

**Weaknesses:**

1. The experiments are limited to environments with a small number of agents (up to five). The effectiveness of MADiTS on datasets with a larger number of agents has not been demonstrated.

2. Given the known issues [1] with SMAC (v1), why were results not included for SMACv2?

3. Unlike DiffStitch, which conditions on both the first and last trajectory segments, the diffusion model in MADiTS generates the subsequent trajectory based solely on the first state, which makes the naming of "Stitch" a little confusing.

4. A key highlight of this paper is the use of integrated gradients to identify underperforming agents. However, this part lacks sufficient analysis. It would be beneficial to quantitatively demonstrate a general correlation between the value of integrated gradients and underperformance degrees across different environments.

5. I have doubts about the extent to which regeneration can improve the overall reward. The multi-agent diffusion model learns from the joint trajectory distribution, meaning it is constrained by the combination of cooperative capabilities existed in the training data. Suppose, in a two-agent environment (agent 0 and agent 1), the training set includes two joint trajectories, $\tau_1$ and $\tau_2$. In $\tau_1$, agent 0 significantly contributes to the team reward, while agent 1’s contribution is minimal. In $\tau_2$, the situation is reversed. Now, suppose the model’s first generation produces a trajectory where agent 0 has high contribution and agent 1 has low contribution. If we successfully use integrated gradients to identify agent 1 and attempt to condition on agent 0’s trajectory to regenerate agent 1’s trajectory, the model should be unable to produce a high-contribution trajectory for agent 1, as the training dataset lacks joint trajectories where both agents have high contributions. Therefore, in this case, MADiTS may not effectively combine individual trajectories to yield better joint trajectories. The authors should discuss similar cases, clarifying the method’s limitations and maybe requirements for the training dataset.

[1] Ellis, Benjamin, et al. "Smacv2: An improved benchmark for cooperative multi-agent reinforcement learning." *Advances in Neural Information Processing Systems* 36 (2024).

Update: Since the authors' responses have adequately addressed my concerns, I recommend accepting the paper.

**Questions:**

See weaknesses.

---

> ### Author Response · Authors · 2024-11-20
> **Response to Reviewer qWY9 (Part I)**
>
> Thank you very much for carefully reviewing our paper and providing constructive comments and suggestions. In response, we have conducted additional experiments and clarified the points of confusion. The detailed response is presented as follows:
>
> **Q1: The effectiveness of MADiTS on datasets with a larger number of agents.**
>
> A1: MADiTS demonstrates enhanced sample efficiency across various methods, even as the number of agents increases. To validate its effectiveness, we conducted experiments on the 12m map from SMAC, with 12 marine agents and 12 marine enemies. The results, shown in the table below, indicate that MADiTS outperforms the baseline method MA-MBTS and achieves competitive performance compared to MADiff, highlighting its broad applicability. Addressing scalability in large-scale MARL settings with hundreds or thousands of agents[1], possibly through techniques like agent grouping, is a promising direction for future work.
>
> | Envs | Algs | Balanced | Original |  | MA-MBTS |  | MADiff |  | MADiTS |  |
> | --- | --- | --- | --- | --- | --- | --- | --- | --- | --- | --- |
> |  |  | exp | exp-m | exp-s | exp-m | exp-s | exp-m | exp-s | exp-m | exp-s |
> | 12m | BC | 0.95 ± 0.03 | 0.83 ± 0.11 | 0.63 ± 0.28 | 0.81 ± 0.11 | 0.56 ± 0.22 | 0.85 ± 0.12 | **0.65 ± 0.21** | **0.88 ± 0.07** | **0.65 ± 0.12** |
> |  | OMIGA | 0.97 ± 0.03 | 0.94 ± 0.02 | 0.85 ± 0.02 | 0.95 ± 0.03 | 0.85 ± 0.03 | **0.96 ± 0.03** | 0.86 ± 0.04 | 0.95 ± 0.02 | **0.87 ± 0.02** |
> |  | CFCQL | 0.90 ± 0.04 | 0.78 ± 0.09 | 0.50 ± 0.11 | 0.70 ± 0.11 | 0.46 ± 0.10 | 0.79 ± 0.07 | 0.53 ± 0.09 | **0.80 ± 0.05**  | **0.63 ± 0.08** |
> | Average |  | 0.94 | 0.85 | 0.66 | 0.82 | 0.62 | 0.86 | 0.68 | **0.87** | **0.71** |
>
> **Q2: Results on SMACv2**
>
> A2: Our method enhances sample efficiency across various offline MARL approaches and benchmarks. In addition to the commonly used SMAC and MPE, we conducted experiments on the terran_5_vs_5 and zerg_5_vs_5 maps from SMACv2. As shown in the following table, MADiTS consistently improves the sample efficiency of algorithms such as BC, OMIGA, and CFCQL under different data imbalance settings, demonstrating its general applicability.
>
> | Envs | Algs | Balanced | Original |  | MA-MBTS |  | MADiff |  | MADiTS |  |
> | --- | --- | --- | --- | --- | --- | --- | --- | --- | --- | --- |
> |  |  | exp | exp-m | exp-s | exp-m | exp-s | exp-m | exp-s | exp-m | exp-s |
> | terran_5_vs_5 | BC | 0.55 ± 0.08 | 0.46 ± 0.02 | 0.38 ± 0.27 | 0.46 ± 0.05 | 0.41 ± 0.13 | 0.51 ± 0.03 | 0.42 ± 0.08 | **0.54 ± 0.08** | **0.46 ± 0.09** |
> |  | OMIGA | 0.59 ± 0.06 | 0.55 ± 0.15 | 0.54 ± 0.11 | 0.58 ± 0.08 | 0.56 ± 0.04 | 0.61 ± 0.06 | 0.57 ± 0.04 | **0.70 ± 0.06** | **0.61 ± 0.08** |
> |  | CFCQL | 0.65 ± 0.09 | 0.54 ± 0.12 | 0.49 ± 0.09 | 0.54 ± 0.10 | 0.49 ± 0.19 | 0.53 ± 0.06 | 0.50 ± 0.07 | **0.55 ± 0.11** | **0.52 ± 0.06** |
> | zerg_5_vs_5 | BC | 0.32 ± 0.16 | 0.30 ± 0.19 | 0.27 ± 0.13 | 0.28 ± 0.08 | 0.29 ± 0.13 | 0.36 ± 0.07 | 0.33 ± 0.08 | **0.40 ± 0.06** | **0.38 ± 0.06** |
> |  | OMIGA | 0.44 ± 0.09 | 0.34 ± 0.05 | 0.31 ± 0.09 | 0.36 ± 0.11 | 0.34 ± 0.07 | 0.36 ± 0.13 | 0.33 ± 0.11 | **0.38 ± 0.07** | **0.37 ± 0.13** |
> |  | CFCQL | 0.55 ± 0.07 | 0.49 ± 0.11 | 0.25 ± 0.09 | 0.47 ± 0.14 | 0.31 ± 0.09 | 0.46 ± 0.07 | 0.34 ± 0.10 | **0.50 ± 0.08** | **0.42 ± 0.15** |
> | Average |  | 0.51 | 0.44 | 0.37 | 0.45 | 0.40 | 0.47 | 0.42 | **0.51** | **0.46** |
>
> **Q3: Confusion of the term “stitching”.**
>
> A3: In the context of our work, *stitching* refers to the **head-to-tail concatenation of two trajectory segments** to create a complete trajectory. This definition aligns with certain interpretations in the literature, where *stitching* is used metaphorically to describe various operations on trajectories. Existing works in the Offline RL domain employing *stitching* primarily focus on three main approaches:
>
> 1. **Head-to-tail concatenation of two trajectory segments** (e.g., SSD[2]): In this approach, each trajectory segment is generated to ensure high quality, and subsequent segments are initiated from the endpoints of the preceding ones, achieving head-to-tail concatenation.
> 2. **Creating new transitions or sub-trajectories as bridges between existing trajectories** (e.g., DiffStitch[3], MBTS[4]): This involves generating a "bridge" to connect trajectory A and trajectory B, resulting in a new optimal trajectory C, structured as A + bridge + B.
> 3. **Describing the ability to learn optimal policies from suboptimal data** (e.g., QDT[5], EDT[6], WT[7]): Unlike the previous two, this interpretation uses *stitching* as a metaphor for the ability to extract optimal policies from suboptimal trajectories, without explicitly concatenating or modifying trajectories.
>
> Thank you for bringing this point to our attention. We have further clarified the definition and application of stitching in the revised manuscript.

---

> > ### Author Response · Authors · 2024-11-20
> > **Response to Reviewer qWY9 (Part II)**
> >
> > **Q4: Analysis on effects of integrated gradient to identify underperforming agents.**
> >
> > A4: Integrated Gradients (IG)[8] is a widely used neural network interpretability method that uses path integrals of gradients along the path between a baseline input and the actual input to determine how each feature influences the model’s output. This approach tracks the change in predictions from $F(b)$ to $F(x)$ as the input moves from the baseline $b$ to the actual input $x$. We utilize IG’s attribution capabilities to identify underperforming agents in joint trajectory segments. Specifically, in MADiTS, we employ PathIG to estimate each agent’s contribution at every timestep using a trainable reward function. These contributions are aggregated to identify underperforming agents. For instance, in a 3-agent environment (agents 0, 1, and 2), if agent 2 consistently underperforms and provides minimal contributions to the team’s overall performance, PathIG assigns low contribution scores to agent 2 across multiple timesteps. Consequently, the aggregated ranking for agent 2 will be lower than those for agents 0 and 1, clearly identifying agent 2 as underperforming.
> >
> > To quantitatively evaluate the accuracy of IG-based agent rankings, we analyzed the **Kendall correlation coefficient**[9] between the estimated contribution values from IG and the ground-truth contributions in the Cooperative Navigation task. In this task, the proximity of agents to landmarks determines the overall return. Using the true reward function of the environment (available only for analysis), we calculated each agent’s actual contribution at every timestep as the ground truth. The Kendall correlation coefficient was then computed to assess the alignment between IG-derived rankings and ground-truth rankings for each trajectory.
> >
> > As shown in the distribution of Kendall correlation coefficients, most values are **positive**, indicating that IG aligns well with the ground-truth contributions. This demonstrates that our PathIG-based offline credit assignment method achieves a **high degree of accuracy**. Additional details on the results and methodology are provided in Figure 6 and Appendix H.5. We deeply appreciate your feedback, which has helped enhance the clarity and presentation of our method.
> >
> > **Q5: The extent to which regeneration can improve the overall reward.**
> >
> > A: Trajectory regeneration can enhance overall rewards in MARL but is inherently influenced by the quality of the collected training data, a common challenge faced by popular generative models [10]. Our method leverages the robust data distribution modeling capabilities of diffusion models to sample and generate high-quality cooperative trajectory segments for data augmentation, even when these segments are scarce in the original dataset. By addressing dataset imbalances, MADiTS demonstrates significant improvements, as shown in our experiments. Specifically, regeneration reduces the average number of underperforming agents from 0.38 to 0.09 in the CN exp-m dataset, underscoring the efficiency of our approach.
> >
> > Regarding the extreme scenario raised by the reviewer, where the dataset entirely lacks high-quality cooperative segments, generating such Out-of-Distribution (OOD) segments remains a notable challenge in data generation [11]. One potential direction to overcome this issue is integrating external knowledge, such as leveraging Large Language Models (LLMs) in multi-agent RL [12]. We deeply appreciate the reviewer’s insightful question and plan to explore this promising avenue in future work.

---

> > > ### Author Response · Authors · 2024-11-20
> > > **Response to Reviewer qWY9 (Part III)**
> > >
> > > **References:**
> > >
> > > [1] Zheng, L., et al. Magent: A many-agent reinforcement learning platform for artificial collective intelligence. In *Proceedings of the AAAI conference on artificial intelligence*. 2018, 32(1).
> > >
> > > [2] Kim, S., et al. Stitching Sub-trajectories with Conditional Diffusion Model for Goal-Conditioned Offline RL. In *Proceedings of the AAAI Conference on Artificial Intelligence*. 2024, 38(12): 13160-13167.
> > >
> > > [3] Li, G., et al. Diffstitch: Boosting offline reinforcement learning with diffusion-based trajectory stitching. In *International Conference on Machine Learning*, pp. 28597–28609, 2024.
> > >
> > > [4] Hepburn, C. A., et al. Model-based trajectory stitching for improved offline reinforcement learning. *arXiv preprint arXiv:2211.11603*, 2022.
> > >
> > > [5] Yamagata, T., et al. Q-learning decision transformer: Leveraging dynamic programming for conditional sequence modelling in offline RL. In *International Conference on Machine Learning*. 2023: 38989-39007.
> > >
> > > [6] Wu, Y. H., et al. Elastic decision transformer. In *Advances in Neural Information Processing Systems*, 2023.
> > >
> > > [7] Badrinath, A., et al. Waypoint transformer: Reinforcement learning via supervised learning with intermediate targets. In *Advances in Neural Information Processing Systems*, 2023.
> > >
> > > [8] Sundararajan, M., et al. Axiomatic attribution for deep networks. In *International Conference on Machine Learning*, pp. 3319–3328, 2017.
> > >
> > > [9] Kendall, M. G., et al. The treatment of ties in ranking problems. *Biometrika*, 33(3):239–251, 1945.
> > >
> > > [10] Cao, H., et al. A survey on generative diffusion models. In *IEEE Transactions on Knowledge and Data Engineering*, 2024.
> > >
> > > [11] Zhu, Y., et al. Unseen Image Synthesis with Diffusion Models. *arXiv preprint arXiv:2310.09213*, 2023.
> > >
> > > [12] Sun, C., et al. LLM-based Multi-Agent Reinforcement Learning: Current and Future Directions. *arXiv preprint arXiv:2405.11106*, 2024.

---

> > > > ### Comment · Reviewer_qWY9 · 2024-11-22
> > > >
> > > > I appreciate the authors' detailed responses and the additional experimental results. The analysis of integrated gradients in multi-agent learning is well-founded. I suggest that the authors include a discussion on the potential limitations related to Q5.
> > > >
> > > > Since the authors' responses have adequately addressed my concerns, I recommend accepting the paper.

---

> > > > > ### Author Response · Authors · 2024-11-26
> > > > >
> > > > > Thank you for your valuable contribution to our community. We appreciate your positive feedback on our paper and are grateful for your insightful suggestions regarding the regeneration problem, which have greatly enhanced our work. We will incorporate these suggestions into the discussion accordingly.

---

### Official Review · Reviewer_onPK · 2024-11-03

**Soundness:** 3
**Presentation:** 3
**Contribution:** 3
**Rating:** 6
**Confidence:** 4

**Summary:**

The paper proposes the MADiTS method to address the imbalance of offline datasets in both temporal and spatial dimensions. Specifically, it uses diffusion models to generate trajectory segments, then employs bidirectional environmental dynamics constraint to ensure the consistency of trajectories with environmental dynamics, and finally utilizes offline credit assignment technique to identify and optimize the behavior of underperforming agents in the generated segments.

**Strengths:**

1.	The overall structure of the paper is complete, the content is substantial, and the experimental evidence is convincing.
2.	Generating trajectories using diffusion models and ensuring the generated segments conform to environmental dynamics with bidirectional dynamic constraints is similar to learning a world model and using it to verify the rationality of trajectories.
3.	As a data augmentation method, MADiTS is compatible with any offline MARL algorithm and has good versatility.

**Weaknesses:**

1.	The paper uses several certain threshold hyperparameters (environment-independent), which may make the method too sensitive to the setting of hyperparameters. Moreover, the paper does not conduct experimental analysis on methods with different hyperparameters.
2.	MADiTS involves multiple models and steps, which may lead to computational complexity. Additionally, the paper does not include experimental analysis on this aspect.

**Questions:**

1.	How does the proposed method address the imbalance of the dataset in terms of time and space? For example, which technique solves the time imbalance problem, and which solves the spatial imbalance problem?
2.	How are environment-independent hyperparameters like $\delta_{recon}$ determined? What effects do different values have?

---

> ### Author Response · Authors · 2024-11-20
> **Response to Reviewer onPK**
>
> Thank you for your inspiring and thoughtful reviews. We have prepared the following experimental results and comments for your proposed weakness and questions, and we hope they can relieve your concern:
>
> **Q1: Experimental analysis on methods with different hyperparameters and how they are determined.**
>
> A1: We use grid search to select these hyperparameters, which is popular in hyperparameters selection[1]. Take $\delta_{recon}$ as an example, which determines the strictness of the dynamics constraint. We first specify a list of candidate values, such as [0.003, 0.005, 0.01, 0.02, 0.05], and then evaluate the filtering strictness on a small-scale dataset. If $\delta_{recon}$ is too small, stitching a complete trajectory will take a long time. Conversely, if $\delta_{recon}$ is too large, no generated trajectories will be discarded. Details of experimental results for other key hyperparameters are included in Appendix H.3 of the revised paper.
>
> **Q2: Regarding to computational complexity.**
>
> A2: Our method achieves computational complexity comparable to other baseline approaches. To demonstrate this, we provide a summary of the computational costs for MADiTS and baseline methods on the Cooperative Navigation task. The experiments were conducted on servers equipped with GeForce RTX 2080 Ti GPUs. As shown in the table below, MADiTS maintains computational complexity similar to other methods, staying within an acceptable range while ensuring competitive performance.
>
> | **Method** | **Model Training (GPU Hours)** | **Trajectory Processing (GPU Hours)** | **Total (GPU Hours)** |
> | --- | --- | --- | --- |
> | **MADiTS** | 36 | 4 | 40 |
> | **MADiff** | 36 | - | 36 |
> | **MA-MBTS** | - | 48 | 48 |
>
> **Q3: How does the proposed method address the imbalance of the dataset in terms of time and space?**
>
> A3: We leverage a diffusion model to seamlessly stitch trajectory segments across different timesteps, effectively addressing **temporal imbalances**. To tackle **spatial imbalances**, we employ Integrated Gradients to identify these imbalances and optimize them by regenerating data through the diffusion model. We apologize for the earlier unclear explanation and have revised the introduction for clarity in the updated manuscript.
>
> **Reference:**
>
> [1] Liashchynskyi P., et al. Grid search, random search, genetic algorithm: a big comparison for NAS. *arXiv preprint arXiv:1912.06059*, 2019.

---

> ### Author Response · Authors · 2024-11-24
> **Dear Reviewer onPK, are our responses address your questions?**
>
> Dear Reviewer onPK:
>
> We thank you again for your comments and hope our responses could address your questions. As the response system will end in four days, please let us know if we missed anything. More questions on our paper are always welcomed.
>
> Sincerely yours,
>
> Authors of Paper10913

---

### Official Review · Reviewer_Af9P · 2024-11-04

**Soundness:** 3
**Presentation:** 3
**Contribution:** 3
**Rating:** 6
**Confidence:** 4

**Summary:**

The paper investigates the use of diffusion models for offline multi-agent reinforcement learning (MARL), extending certain diffusion-based approaches originally developed for single-agent offline RL to a multi-agent setting. Specifically, the authors combine Denoising Diffusion Probabilistic Models (DDPM), stitching techniques (to merge high-reward and low-reward data points and address data imbalance), and Integrated Gradients (IG) for identifying underperforming agents. This results in a data augmentation process that can enhance offline datasets and other MARL algorithms. Experiments are conducted using MPE and SMACv1, with comparisons to BC and OMIGA.

**Strengths:**

- The diffusion approach makes sense. Using a diffusion model to enrich offline datasets could support several other MARL algorithms.
- The employed techniques align with recent advancements in diffusion-based RL.
- The paper is well-written and easy to follow.

**Weaknesses:**

1. The techniques seem to be a straightforward extension from diffusion-based single-agent RL.
2. Most techniques, except those in Section 4.2, appear to be directly adapted from single-agent settings, with single-agent states and actions replaced by global observations and actions.
3. There is already existing work on diffusion models for offline MARL. For instance, how does this approach compare to MADiff [1], which also develops diffusion models to enhance offline datasets in offline MARL? Notably, MADiff compares its methods against several MARL baselines, which seems lacking in this work.
4. In addition to MPE and SMACv1, more challenging MARL tasks like MAMuJoCo and SMACv2 should be considered for comparisons.
5. It’s unclear how this approach compares to other data-augmentation techniques in offline MARL. Additional citations, discussions, and experimental comparisons would be needed.


**Reference:**
[1] Zhu, Z., Liu, M., Mao, L., Kang, B., Xu, M., Yu, Y., Ermon, S., & Zhang, W. (2023). MADiff: Offline Multi-agent Learning with Diffusion Models. arXiv preprint arXiv:2305.17330.

**Questions:**

- How does your approach compare to MADiff?
- Can these methods be applied to cooperative-competitive settings?
- Would the approach work effectively with balanced but bad datasets? What would happen if the data were balanced but consisted primarily of transitions with low rewards?

**Details Of Ethics Concerns:**

There might be some ethical concerns:
- Diffusion models trained on offline datasets may inherit biases from the original data, particularly if it is imbalanced or contains biases toward certain actions or outcomes. This could result in the reinforcement of unintended behaviors, potentially harmful in applications like autonomous driving, healthcare, or finance.
- If the diffusion model generates synthetic data that diverges from real-world scenarios, this could lead to decisions that are unreliable in practice.
- The diffusion model could produce agent interactions that are difficult to predict or manage, raising questions about accountability if something goes wrong.

---

> ### Author Response · Authors · 2024-11-20
> **Response to Reviewer Af9P (Part I)**
>
> Thank you very much for carefully reviewing our paper and providing constructive comments and suggestions, which have helped improve the work a lot. Our response is presented as follows:
>
> **Q1: a straightforward extension from diffusion-based single-agent RL? with single-agent states and actions replaced by global observations and actions.**
>
> A1: Our method is the first to integrate a diffusion model for data generation with Trajectory Stitching, enhancing coordination in multi-agent reinforcement learning (MARL) and addressing challenges beyond single-agent RL[1]. While both approaches aim to improve sample efficiency, MARL presents unique difficulties due to complex agent interactions. Specifically, we leverage the diffusion model's generative capabilities to handle two critical types of trajectory data imbalances: **temporal**, concerning relationships across timesteps, and **spatial**, focusing on interactions among agents. These imbalances complicate data representation and utilization, hindering MARL progress.
>
> To address these issues, we propose two innovations. A bidirectional dynamics constraint ensures generated trajectories align with environmental dynamics. An offline credit assignment technique identifies and optimizes underperforming agents within trajectory segments. Experimental results show that MADiTS achieves superior sample efficiency across diverse MARL tasks, effectively tackling data imbalance and coordination challenges.
>
> **Q2: How does this approach compare to MADiff.**
>
> A2: Our method MADiTS differs from MADiff in addressing sample efficiency challenges in MARL. MADiTS focuses on data augmentation by combining a diffusion model with dynamic detection and credit assignment techniques for offline MARL policy training. In contrast, MADiff primarily uses a diffusion model for trajectory prediction in action planning. Specifically, MADiTS specifically targets data imbalance issues in MARL, while MADiff aims to enhance learning efficiency in standard offline MARL settings. To emphasize these distinctions, we compare MADiTS with MADiff across various environments. The results, summarized in the table below, show that MADiTS outperforms MADiff and other baselines, demonstrating its superior effectiveness in handling data imbalance challenges. We also include performance of other MARL algorithms on the imbalanced dataset, with detailed results presented in the *Original* column.
>
> | Envs | Algs | Original |  | MADiff |  | MADiTS |  |
> | --- | --- | --- | --- | --- | --- | --- | --- |
> |  |  | exp-m | exp-s | exp-m | exp-s | exp-m | exp-s |
> | CN | BC | 17.27 ± 3.66 | 9.03 ± 3.40 | 40.44 ± 3.75 | 30.24 ± 4.82 | **43.44 ± 6.11** | **37.82 ± 4.45** |
> |  | OMIGA | -2.27 ± 57.07 | -11.60 ± 58.90 | 19.26 ± 67.68 | 7.27 ± 66.89 | **23.02 ± 69.69** | **22.91 ± 44.94** |
> |  | CFCQL | -33.40 ± 44.28 | -56.44 ± 40.38 | 30.08 ± 12.97 | 21.88 ± 10.71 | **39.57 ± 16.14** | **28.60 ± 20.94** |
> | PP | BC | 48.49 ± 4.69 | 49.21 ± 3.90 | 49.55 ± 6.50 | 49.93 ± 3.74 | **54.85 ± 4.23** | **55.50 ± 4.28** |
> |  | OMIGA | 37.90 ± 25.34 | 26.73 ± 40.67 | 47.48 ± 20.97 | 58.44 ± 4.76 | **63.02 ± 3.40** | **63.71 ± 5.67** |
> |  | CFCQL | 45.03 ± 4.62 | 28.88 ± 6.29 | 45.50 ± 6.76 | 30.04 ± 7.02 | **47.38 ± 3.74** | **32.25 ± 10.98** |
> | World | BC | 47.29 ± 3.00 | 48.30 ± 5.96 | 50.79 ± 2.62 | 51.55 ± 4.96 | **54.25 ± 4.35** | **52.84 ± 5.29** |
> |  | OMIGA | 54.23 ± 6.20 | 52.92 ± 5.64 | 49.55 ± 24.99 | 48.26 ± 24.75 | **57.35 ± 5.89** | **58.59 ± 9.32** |
> |  | CFCQL | 28.59 ± 1.90 | 18.14 ± 18.16 | 28.63 ± 10.14 | 34.35 ± 6.91 | **29.62 ± 10.91** | **39.36 ± 1.91** |
> | Average |  | 27.01 | 18.35 | 40.14 | 36.88 | **45.83** | **43.50** |
> | 2m_vs_1z | BC | 0.03 ± 0.06 | 0.03 ± 0.06 | 0.30 ± 0.18 | 0.30 ± 0.18 | **0.35 ± 0.20** | **0.35 ± 0.20** |
> |  | OMIGA | 0.59 ± 0.28 | 0.59 ± 0.28 | 0.97 ± 0.04 | 0.97 ± 0.04 | **0.98 ± 0.02** | **0.98 ± 0.02** |
> |  | CFCQL | 0.72 ± 0.24 | 0.72 ± 0.24 | 0.92 ± 0.05 | 0.92 ± 0.05 | **0.94 ± 0.05** | **0.94 ± 0.05** |
> | 3m | BC | 0.42 ± 0.05 | 0.38 ± 0.20 | 0.44 ± 0.18 | 0.36 ± 0.15 | **0.51 ± 0.19** | **0.43 ± 0.18** |
> |  | OMIGA | 0.93 ± 0.05 | 0.90 ± 0.03 | **1.00 ± 0.00** | 0.94 ± 0.03 | **1.00 ± 0.00** | **0.96 ± 0.03** |
> |  | CFCQL | 0.90 ± 0.06 | 0.80 ± 0.08 | 0.89 ± 0.05 | 0.81 ± 0.09 | **0.94 ± 0.07** | **0.89 ± 0.06** |
> | 2s3z | BC | 0.73 ± 0.11 | 0.68 ± 0.14 | 0.75 ± 0.06 | 0.67 ± 0.05 | **0.78 ± 0.06** | **0.69 ± 0.26** |
> |  | OMIGA | 0.86 ± 0.16 | 0.67 ± 0.05 | **1.00 ± 0.00** | 0.71 ± 0.15 | **1.00 ± 0.00** | **0.76 ± 0.19** |
> |  | CFCQL | 0.73 ± 0.15 | 0.66 ± 0.14 | 0.80 ± 0.09 | 0.63 ± 0.15 | **0.92 ± 0.02** | **0.68 ± 0.26** |
> | Average |  | 0.65 | 0.60 | 0.78 | 0.70 | **0.82** | **0.74** |

---

> > ### Author Response · Authors · 2024-11-20
> > **Response to Reviewer Af9P (Part II)**
> >
> > **Q3: Results on more challenging MARL tasks like MAMuJoCo and SMACv2**
> >
> > A3: Our method enhances sample efficiency across a range of offline MARL approaches and benchmarks. Beyond the commonly used SMAC and MPE environments, we conducted experiments on **terran_5_vs_5** and **zerg_5_vs_5** maps from SMACv2[2] and the **4ant** environment from MAMuJoCo[3]. As shown in the table below, MADiTS consistently improves the sample efficiency of algorithms such as BC, OMIGA, and CFCQL under varying data imbalance settings. These results highlight its robustness and general applicability.
> >
> > | Envs | Algs | Balanced | Original |  | MA-MBTS |  | MADiff |  | MADiTS |  |
> > | --- | --- | --- | --- | --- | --- | --- | --- | --- | --- | --- |
> > |  |  | exp | exp-m | exp-s | exp-m | exp-s | exp-m | exp-s | exp-m | exp-s |
> > | terran_5_vs_5 | BC | 0.55 ± 0.08 | 0.46 ± 0.02 | 0.38 ± 0.27 | 0.46 ± 0.05 | 0.41 ± 0.13 | 0.51 ± 0.03 | 0.42 ± 0.08 | **0.54 ± 0.08** | **0.46 ± 0.09** |
> > |  | OMIGA | 0.59 ± 0.06 | 0.55 ± 0.15 | 0.54 ± 0.11 | 0.58 ± 0.08 | 0.56 ± 0.04 | 0.61 ± 0.06 | 0.57 ± 0.04 | **0.70 ± 0.06** | **0.61 ± 0.08** |
> > |  | CFCQL | 0.65 ± 0.09 | 0.54 ± 0.12 | 0.49 ± 0.09 | 0.54 ± 0.10 | 0.49 ± 0.19 | 0.53 ± 0.06 | 0.50 ± 0.07 | **0.55 ± 0.11** | **0.52 ± 0.06** |
> > | zerg_5_vs_5 | BC | 0.32 ± 0.16 | 0.30 ± 0.19 | 0.27 ± 0.13 | 0.28 ± 0.08 | 0.29 ± 0.13 | 0.36 ± 0.07 | 0.33 ± 0.08 | **0.40 ± 0.06** | **0.38 ± 0.06** |
> > |  | OMIGA | 0.44 ± 0.09 | 0.34 ± 0.05 | 0.31 ± 0.09 | 0.36 ± 0.11 | 0.34 ± 0.07 | 0.36 ± 0.13 | 0.33 ± 0.11 | **0.38 ± 0.07** | **0.37 ± 0.13** |
> > |  | CFCQL | 0.55 ± 0.07 | 0.49 ± 0.11 | 0.25 ± 0.09 | 0.47 ± 0.14 | 0.31 ± 0.09 | 0.46 ± 0.07 | 0.34 ± 0.10 | **0.50 ± 0.08** | **0.42 ± 0.15** |
> > | Average |  | 0.51 | 0.44 | 0.37 | 0.45 | 0.40 | 0.47 | 0.42 | **0.51** | **0.46** |
> > | 4ant | BC | 39.43 ± 6.09 | 25.31 ± 8.15 | 10.48 ± 12.58 | 19.75 ± 3.68 | 10.58 ± 3.38 | 28.30 ± 8.60 | 20.29 ± 6.57 | **31.66 ± 5.10** | **21.89 ± 6.23** |
> > |  | OMIGA | 52.30 ± 5.71 | 33.12 ± 10.60 | 31.76 ± 10.99 | 40.00 ± 8.72 | 30.61 ± 8.46 | 40.50 ± 6.74 | 34.35 ± 4.47 | **48.92 ± 6.57** | **43.09 ± 8.15** |
> > |  | CFCQL | 58.87 ± 6.94 | 38.78 ± 7.85 | 7.10 ± 9.85 | 32.41 ± 5.93 | 6.37 ± 5.79 | 41.98 ± 7.12 | 24.03 ± 8.07 | **48.43 ± 5.91** | **35.28 ± 6.86** |
> > | Average |  | 50.20 | 32.40 | 16.44 | 30.72 | 15.85 | 36.92 | 26.22 | **43.00** | **33.42** |

---

> > > ### Author Response · Authors · 2024-11-20
> > > **Response to Reviewer Af9P (Part III)**
> > >
> > > **Q4: Additional citations, discussions, and experimental comparisons to other data-augmentation techniques in offline MARL**
> > >
> > > A4: Our method addresses sample efficiency issues by using diffusion models to perform trajectory stitching for data augmentation in temporally and spatially imbalanced datasets. This approach enables offline MARL algorithms to achieve better performance by learning from the enhanced dataset. Notably, this is an unexplored area in existing research and is investigated for the first time in this paper, setting it apart from the mentioned works.
> > >
> > > - SIT[4] is an offline MARL method that focuses on learning an effective joint policy from agent-wise imbalanced datasets, primarily addressing spatial imbalance. It achieves this through an attention-based reward decomposition network for offline credit assignment, which identifies high-quality individual trajectories for sharing among agents, and a graph attention network for conservative policy training. In contrast, our method MADiTS prioritizes improving sample efficiency through a data-oriented augmentation pipeline. By employing a diffusion model-based trajectory stitching mechanism, MADiTS enhances dataset quality to address both temporal and spatial imbalances. Unlike SIT, which directly learns policies from imbalanced datasets, MADiTS generates augmented datasets that can be flexibly used by any offline MARL algorithm, providing enhanced flexibility and modularity.
> > > - DOM2[5] is a diffusion model-based offline MARL algorithm that aims to enhance policy expressiveness and diversity. It achieves notable improvements in performance, generalization, and data efficiency through an accelerated solver for diffusion-based policy construction and a policy regularizer, while also scaling up dataset size to boost policy learning. DOM2 excels on balanced datasets, outperforming traditional conservatism-based offline MARL methods. In contrast, MADiTS targets the challenges posed by imbalanced datasets. By addressing temporal and spatial imbalances, MADiTS improves dataset quality, enabling other offline MARL algorithms to achieve better performance and demonstrating its effectiveness in handling imbalanced scenarios.
> > > - MADiff[6] is a diffusion model-based offline MARL algorithm designed to predict future joint actions for decision-making by modeling teammate behaviors. It uses an attention-based diffusion model to capture joint observation sequences and infer actions for planning, achieving strong performance on balanced datasets in standard offline MARL settings. In contrast, MADiTS specifically targets the challenges of data imbalance in offline MARL, focusing on improving sample efficiency through innovative data augmentation techniques. While MADiff enhances learning efficiency in balanced scenarios, MADiTS addresses the unique difficulties posed by imbalanced datasets. To highlight these distinctions, we compared the performance of MADiff above, extended for data augmentation, with MADiTS across various environments. The results demonstrate MADiTS’s superior effectiveness in tackling data imbalance challenges.
> > >
> > > We provide a comparison of our method with MADiff across various environments in Table 1 and Table 4 in the revised manuscript. Additionally, key differences with other related methods on relevant topics are discussed in detail in Appendix B.
> > >
> > > **Q5: Can these methods be applied to cooperative-competitive settings?**
> > >
> > > A5: Our paper mainly addresses the cooperative multi-agent reinforcement learning (MARL) problem[7], a widely studied setup where all agents share a global reward. Through extensive experiments across various offline MARL environments, we demonstrate that MADiTS significantly improves MARL performance. While our current focus is on cooperative settings, the method can be naturally extended to cooperative-competitive scenarios by equipping each team with its own MADiTS model. In this setup, teams can independently learn model parameters using tailored buffers for training. We thank the reviewer for highlighting this intriguing direction and plan to explore it further in future work.

---

> > > > ### Author Response · Authors · 2024-11-20
> > > > **Response to Reviewer Af9P (Part IV)**
> > > >
> > > > **Q6: Results on balanced but bad datasets.**
> > > >
> > > > A6: Our method shows significant improvements even on balanced but low-quality datasets. Specifically, we use the multi-agent offline dataset in MPE with continuous action spaces, as constructed in OMAR[8] for data augmentation. This dataset includes trajectories generated by fully trained expert policies, medium-performing policies, and random policies in an online environment. As demonstrated in the table below, MADiTS consistently enhances learning efficiency across expert, medium, md-replay, and random data settings, showcasing its versatility and robustness across diverse scenarios.
> > > >
> > > > | Env | Dataset | Original | MADiTS |
> > > > | --- | --- | --- | --- |
> > > > | CN | expert | 100.42 ± 5.67 | 127.91 ± 5.63 |
> > > > |  | medium | 80.46 ± 5.05 | 91.31 ± 2.24 |
> > > > |  | md-replay | 32.17 ± 4.51 | 84.47 ± 8.04 |
> > > > |  | random | 2.69 ± 4.24 | 22.38 ± 4.59 |
> > > > | PP | expert | 77.54 ± 5.73 | 97.04 ± 11.81 |
> > > > |  | medium | 54.14 ± 10.19 | 63.25 ± 10.15 |
> > > > |  | md-replay | -6.43 ± 3.21 | 25.04 ± 11.45 |
> > > > |  | random | -8.94 ± 1.55 | -7.38 ± 2.49 |
> > > > | World | expert | 76.89 ± 23.08 | 138.10 ± 34.01 |
> > > > |  | medium | 54.99 ± 14.27 | 96.27 ± 11.78 |
> > > > |  | md-replay | 19.28 ± 4.06 | 29.46 ± 4.65 |
> > > > |  | random | 6.19 ± 3.08 | 14.01 ± 2.95 |
> > > >
> > > > **Q7: Ethical concerns on the use of diffusion models.**
> > > >
> > > > A7:  Biases in the original offline dataset can propagate into the diffusion model, potentially affecting the quality of the generated trajectories. To mitigate this, our method employs a bidirectional dynamics constraint, ensuring that the generated trajectories remain consistent with the environmental dynamics. Additionally, we integrate an offline credit assignment technique to identify and optimize the performance of underperforming agents within the generated trajectory segments, further enhancing the overall quality and utility of the augmented data.
> > > >
> > > > On one hand, generative models like ChatGPT[9] and SORA[10] rely on large-scale datasets to train architectures such as Transformers or diffusion-based models. These models exhibit exceptional generative capabilities across domains like language and video, aligning with scaling laws that link performance to data size. Recognizing the importance of data, these methods often use autoregressive training or advanced techniques to optimize data fitting. For real-world applications such as autonomous driving[11], healthcare[12], and finance[13], diffusion models require extensive and diverse datasets to ensure robust performance. Recent advancements have highlighted the potential of diffusion models to transform these domains.
> > > >
> > > > On the other hand, despite their capabilities, these methods face challenges such as unreliable or unrepresentative data. To ensure reliability in real-world applications, techniques like human-in-the-loop testing[14] and risk control mechanisms[15] are crucial.
> > > >
> > > > To address the issue of synthetic data deviating from real-world distributions, our method MADiTS  introduces a bidirectional dynamics constraint to align generated trajectories with environmental dynamics. Moreover, the offline credit assignment technique enhances robustness by identifying and improving underperforming agents in generated segments. Experimental results validate the effectiveness of MADiTS in overcoming these challenges.
> > > >
> > > > We appreciate your insightful concerns regarding the use of diffusion models with offline datasets and look forward to continued discussions on advancing this field.

---

> > > > > ### Author Response · Authors · 2024-11-20
> > > > > **Response to Reviewer Af9P (Part V)**
> > > > >
> > > > > **References:**
> > > > >
> > > > > [1] Lu, C., et al. Synthetic experience replay. In *Advances in Neural Information Processing Systems*, 2023.
> > > > >
> > > > > [2] Ellis, B., et al. SMACv2: An improved benchmark for cooperative multi-agent reinforcement learning. In *Advances in Neural Information Processing Systems*, 2023.
> > > > >
> > > > > [3] Peng, B., et al. FACMAC: Factored multi-agent centralised policy gradients. In *Advances in Neural Information Processing Systems*, 34:12208–12221, 2021.
> > > > >
> > > > > [4] Tian, Q., et al. Learning from good trajectories in offline multi-agent reinforcement learning. In *Association for the Advancement of Artificial Intelligence*, pp. 11672–11680, 2023.
> > > > >
> > > > > [5] Li, Z., et al. Beyond conservatism: Diffusion policies in offline multi-agent reinforcement learning. *arXiv preprint arXiv:2307.01472*, 2023.
> > > > >
> > > > > [6] Zhu, Z., et al. MADiff: Offline multi-agent learning with diffusion models. *arXiv preprint arXiv:2305.17330*, 2023.
> > > > >
> > > > > [7] Oroojlooy, A., et al. A review of cooperative multi-agent deep reinforcement learning. *Applied Intelligence*, pp. 1–46, 2022.
> > > > >
> > > > > [8] Pan, L., et al. Plan better amid conservatism: Offline multi-agent reinforcement learning with actor rectification. In *International Conference on Machine Learning*, pp. 17221–17237, 2022.
> > > > >
> > > > > [9] OpenAI. ChatGPT: Optimizing language models for dialogue. 2023. URL: https://www.openai.com/chatgpt.
> > > > >
> > > > > [10] Brooks, T., et al. Video generation models as world simulators. 2024. URL: https://openai.com/research/video-generation-models-as-world-simulators.
> > > > >
> > > > > [11] Wang, X., et al. DriveDreamer: Towards real-world-driven world models for autonomous driving. In *European Conference on Computer Vision*, 2024.
> > > > >
> > > > > [12] Kazerouni, A., et al. Diffusion models in medical imaging: A comprehensive survey. *Medical Image Analysis*, 2023, 88: 102846.
> > > > >
> > > > > [13] Wang, Z., et al. A financial time series denoiser based on diffusion models. In *Proceedings of the 5th ACM International Conference on AI in Finance*. 2024: 72–80.
> > > > >
> > > > > [14] Singi, S., et al. Decision making for human-in-the-loop robotic agents via uncertainty-aware reinforcement learning. In *2024 IEEE International Conference on Robotics and Automation (ICRA)*. IEEE, 2024: 7939–7945.
> > > > >
> > > > > [15] Yang, Q., et al. WCSAC: Worst-case soft actor critic for safety-constrained reinforcement learning. In *Proceedings of the AAAI Conference on Artificial Intelligence*. 2021, 35(12): 10639–10646.

---

> ### Author Response · Authors · 2024-11-24
> **Dear Reviewer Af9P, are our responses address your questions?**
>
> Dear Reviewer Af9P:
>
> We thank you again for your comments and hope our responses could address your questions. As the response system will end in four days, please let us know if we missed anything. More questions on our paper are always welcomed.
>
> Sincerely yours,
>
> Authors of Paper10913

---

> ### Comment · Reviewer_Af9P · 2024-11-25
>
> Thank you to the authors for making a serious effort to address my concerns. I believe the additional experiments provide better clarity on the contributions. I am now more positive about the paper and will increase my rating after seeing the authors updating the paper with the additional experiments and discussions.

---

> > ### Author Response · Authors · 2024-11-26
> > **Thank you for your timely response!**
> >
> > We highly appreciate the reviewer’s insightful feedback, which clearly helped us improve our paper a lot. Regarding the additional experiments on more challenging MARL tasks (MAMuJoCo and SMACv2), due to the 10-page limit for the main paper, we have placed the results in **Table 4 of Appendix H.1** in the revised paper. The analysis of these additional experimental results is also highlighted in blue for clarity. We have also **uploaded a new version of manuscript** and added further **discussion on the application to cooperative-competitive settings and concerns on the use of diffusion models** in **Appendix I**, which is also highlighted for clarity.
> >
> > We sincerely appreciate the reviewer find our response useful. If you have any further questions, we would be delighted to discuss them in more detail!

---

> > > ### Comment · Reviewer_Af9P · 2024-11-27
> > >
> > > I thank the authors for the updates. I have increased my rating.
> > >
> > > Best --

---

> > > > ### Author Response · Authors · 2024-11-27
> > > >
> > > > Thank you for sharing your time in evaluating our paper, and supporting our community.

---

### Author Response · Authors · 2024-11-20
**General Response**

We really appreciate your efforts in helping us reflect and improve the paper. In light of the reviewers' comments, we made the following major revision based on the previous manuscript and now submitted as the rebuttal version. The revised parts are highlighted in blue.

1. A new baseline MADiff is included for all environments in **Section 5.2** and **Appendix H.1**. **(for reviewer Af9P and bitr)**
2. Results on more challenging MARL tasks like MAMuJoCo and SMACv2 are included in **Appendix H.1**. **(for reviewer Af9P and qWY9)**
3. Results on datasets with a larger number of agents are included in **Appendix H.1**. **(for reviewer qWY9 and bitr)**
4. Detailed discussion on key differences with other related methods on offline MARL are discussed in **Appendix B**.  **(for reviewer Af9P and bitr)**
5. More related works are included in the revised manuscript in **Section 2**. **(for reviewer bitr)**
6. Computational cost of our method and baselines are discussed in **Appendix E**. **(for reviewer onPK and bitr)**
7. Vague description of “continues until we obtain a satisfactory augmented episode trajectory” is revised in **Abstract** and **Introduction**. **(for reviewer bitr)**
8. Unclear explanation of how we address the imbalance of the dataset in terms of time and space is revised in **Introduction**. **(for reviewer onPK)**
9. The meaning of “stitching” is further explained in Introduction and **Section 4.1**. **(for reviewer qWY9)**
10. Quantitative Analysis of the accuracy and effectiveness of Integrate Gradient is included in **Appendix H.5**. **(for reviewer qWY9)**
11. Further Discussion of the limitations and applicability is included in **Conclusion**. **(for reviewer qWY9)**
12. Illustration of impact of MADiTS on the diversity of the trajectories is included in **Appendix H.6**. **(for reviewer bitr)**

Thank you all again for your time and valuable advice! We will address each reviewer's individual concerns momentarily. Feel free to let us know if you have any more comments or questions.

---

### Meta-Review · Area_Chair_HkWC · 2024-12-21

**Metareview:**

his paper introduces MADiTS, a novel diffusion-based trajectory stitching method for offline Multi-Agent Reinforcement Learning (MARL), addressing temporal and spatial data imbalances. The proposed method effectively generates high-quality trajectories by integrating a diffusion model, bidirectional dynamics constraints, and offline credit assignment to identify and enhance underperforming agents. Strengths include the innovative use of trajectory augmentation, robust experimental results demonstrating state-of-the-art performance across diverse MARL tasks, and clear explanations of its contributions. While computational cost analysis and extensions to large-scale settings are areas for future work, the thorough rebuttal and additional experiments addressing scalability, diversity, and comparisons with baselines like MADiff and DOM2 solidify the paper's contributions. These merits strongly support its acceptance.

**Additional Comments On Reviewer Discussion:**

During the rebuttal period, reviewers raised concerns about scalability, diversity, computational cost, and comparisons with related methods like MADiff, DOM2, and SIT. The authors provided extensive responses, including new experiments on larger MARL settings (e.g., 12m map in SMAC), diversity analysis using ADE metrics, and computational cost evaluations showing competitive performance. They clarified MADiTS's unique contributions and addressed vague descriptors in the original manuscript. These clarifications and the addition of comparative baselines significantly strengthened the paper's case, leading to improved reviewer scores and consensus on its acceptance.

---

### Decision · Program_Chairs · 2025-01-22

Accept (Poster)